# Archaean green-light environments drove the evolution of cyanobacteria's light-harvesting system

Taro Matsuo [1,2] ✉, Kumiko Ito-Miwa [1,2], Yosuke Hoshino[3,4], Yuri I. Fujii[1,5], Satomi Kanno[2], Kazuhiro J. Fujimoto [6,7], Rio Tsuji[7], Shinnosuke Takeda[5], Chieko Onami [5], Chihiro Arai [1], Yoko Yoshiyama [8], Yoshihisa Mino [9], Yuki Kato[1], Takeshi Yanai [6,7], Yuichi Fujita[10], Shinji Masuda [11,12], Takeshi Kakegawa[13] & Hideaki Miyashita[5]

Cyanobacteria induced the great oxidation event around 2.4 billion years ago, probably triggering the rise in aerobic biodiversity. While chlorophylls are universal pigments used by all phototrophic organisms, cyanobacteria use additional pigments called phycobilins for their light-harvesting antennas—phycobilisomes—to absorb light energy at complementary wavelengths to chlorophylls. Nonetheless, an enigma persists: why did cyanobacteria need phycobilisomes? Here, we demonstrate through numerical simulations that the underwater light spectrum during the Archaean era was probably predominantly green owing to oxidized Fe(III) precipitation. The green-light environments, probably shaped by photosynthetic organisms, may have directed their own photosynthetic evolution. Genetic engineering of extant cyanobacteria, simulating past natural selection, suggests that cyanobacteria that acquired a green-specialized phycobilin called phycoerythrobilin could have flourished under green-light environments. Phylogenetic analyses indicate that the common ancestor of modern cyanobacteria embraced all key components of phycobilisomes to establish an intricate energy transfer mechanism towards chlorophylls using green light and thus gained strong selective advantage under green-light conditions. Our findings highlight the co-evolutionary relationship between oxygenic phototrophs and light environments that defined the aquatic landscape of the Archaean Earth and envision the green colour as a sign of the distinct evolutionary stage of inhabited planets.

Carl Sagan described the Earth, as imaged by Voyager 1 at a distance of 6 billion kilometres, as a 'pale blue dot'[1]. This description is a consequence of the Rayleigh scattering of sunlight in the atmosphere, in conjunction with the reflection and scattering across the expanse of the ocean. The pale blue colour, serving as a metaphor, symbolizes the cradle of life. Nevertheless, one might enquire: does only a blue hue of a planet serve as an indicator of its potential to nurture life?

The planetary surface has not only been chemically and physically changed by geological events over 4.5 billion years but has also been moulded by life since its emergence. Cyanobacteria, as pioneering oxygenic photosynthetic organisms, spread across the globe by photolysis of water to generate molecular oxygen using solar radiation. This biological evolution caused a pivotal oxygenation event called the great oxidation event (GOE) around 2.4 billion years ago[2]. The GOE probably played an important role in the promotion of aerobic

**Fig. 1 | Underwater green-light environment after the emergence of cyanobacteria and photoferrotrophs in the Archaean era. a**, Archaean water environment assumption for calculating iron hydroxide concentration. The green-shaded area represents the oxidized region, while the orange dots indicate iron hydroxide particles. The habitats of cyanobacteria (yellow dashed area) and photoferrotrophs (brown dashed area) are inferred to have been separated into oxidized and reduced zones, respectively. Reduced iron from thermal vents at the sea floor was transformed into iron hydroxide by photoferrotrophic and cyanobacterial activities. The white solid vertical column indicates the calculation area, a one-dimensional vertical column with a height of 150 m. **b**, Concentration of iron hydroxide (green), oxygen (red) and reduced iron (blue), with the depth of the pycnocline set at 50 m. **c**, Incident photon flux at the surface water (grey dotted line) and at depths of 5 m (black dashed line) and 20 m (black solid line). The pigment absorption spectra are superimposed: Chl *a* (blue line), PE (green line), PC (orange line) and APC (brown line). Background coloured regions in the figures denote the absorption wavelength ranges of different pigments. **d,e**, Correlation of incident photon flux with photosynthetic pigments at depths of 20 m (**d**) and 5 m (**e**). The colour code is the same as Fig. 1c.

biodiversity[3]. However, it is worth noting that recent research suggests an emergence of aerobic metabolism and thus an oxygenated biosphere before the GOE[4].

Cyanobacteria use a giant and complex light-harvesting system called phycobilisomes as a light-harvesting system and transfer the absorbed energy to photosystem I (PSI) and photosystem II (PSII) for photosynthesis[5]. Phycobilisomes consist mainly of three phycobiliproteins: allophycocyanin (APC), phycocyanin (PC) and phycoerythrin (PE)[6]. The two major phycobilin pigments—phycocyanobilin (PCB) and phycoerythrobilin (PEB)—attach to the phycobiliproteins[7] and can absorb light in the wavelength range 500–650 nm, which is not efficiently absorbed by chlorophyll *a* (Chl *a*). Whereas PC and APC constitute integral components of phycobilisomes, the inclusion of PE occurs sporadically in cyanobacteria[8,9] and is closely related to the various light environments of host habitats[10]. The absorbed light energy is transferred to PSII via PE, PC and APC. Although the energy transfer efficiency is >95% (ref. 11), phycobilisomes are much larger and more complex than other light-harvesting systems such as the light-harvesting complex (LHC) for higher plants[6]. Hence, the rationale behind the exclusive use of phycobilisomes as the light-harvesting system by cyanobacteria remains unclear. Considering that antenna pigments of cyanobacteria are attuned to the light environment of their habitats[10], the light incident to the habitats of ancestral cyanobacteria, referred to as the 'light window', could have played an important role in driving natural selection towards the evolution of phycobilisomes. This co-evolutionary relationship between the light environment and the corresponding photosystem was indeed considered in previous studies[12,13]. Hence, the Archaean underwater light window could have been distinct from the modern continuous light windows over the visible wavelength range (that is, white-light window) and favoured the selection of phycobilisomes. Because cyanobacteria, which probably evolved before the GOE[14–16], traversed a transition from fully reduced to oxidized environments, their habitats and associated light windows conceivably changed along with the gradual oxidation process of the surface of the Earth (Supplementary Discussions 7 and 8).

## Transformation of light window for photosynthetic organisms

We explored the transformation of the light window alongside surface oxidation during the Archaean era, particularly with the rise of photosynthetic organisms. Initially, reduced iron Fe(II) dissolved completely in the reduced aquatic environment[17] (Supplementary Discussions 1, 7 and 8). However, the emergence of cyanobacteria[14–16] and phototrophic Fe(II)-oxidizing bacteria[17,18] in the Archaean era led to the oxidation of Fe(II) (Fig. 1a), forming iron hydroxide precipitates (Fe(OH)₃) and contributing to the formation of banded iron formations (BIFs)[18–21]. Given the vertical structure of open oceans[22,23], iron hydroxide probably spread across cyanobacterial habitats as a result of high eddy diffusivity above the pycnocline.

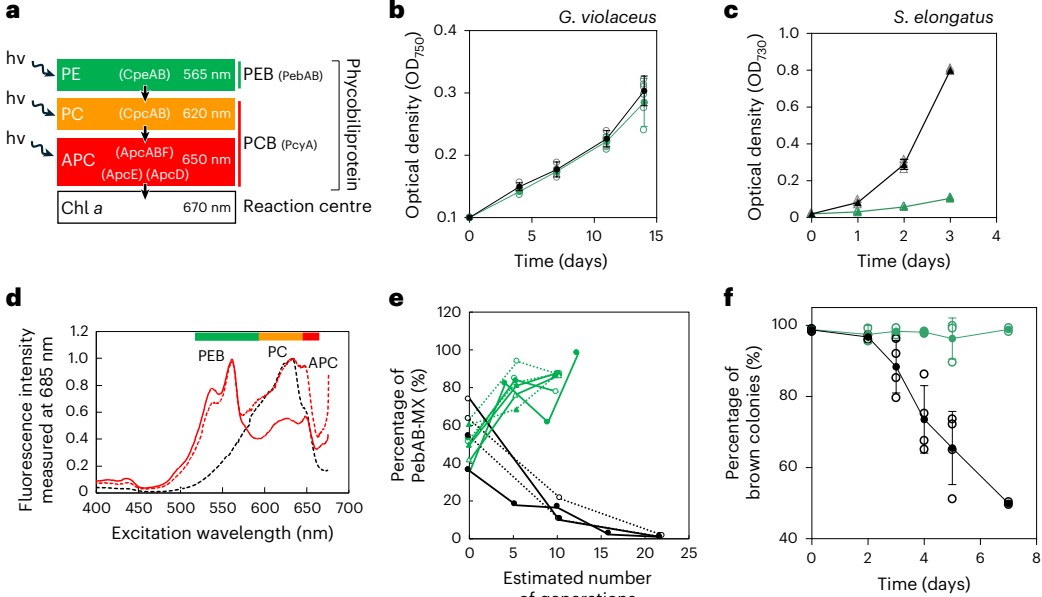

**Fig. 2 | Adaptation of cyanobacteria to the green-light window by PEB.**
**a**, Conceptual diagram for energy transfer from PE to Chl *a* within phycobilisomes, consisting of phycobiliproteins (APC, PC and PE) and conjugated phycobilins (PEB and PCB). Individual components of phycobilisomes and phycobilin synthase were used for the phylogenetic analyses in this study: CpeAB, CpcAB, ApcABF and ApcD for phycobilisomes and PebAB and PcyA for phycobilin synthase. The green, orange and red colours correspond to the absorption wavelengths of PE, PC and APC, respectively. hv represents the energy of a photon with frequency v. **b,c**, Growth of *G. violaceus* PCC 7421 (**b**) and *S. elongatus* PCC 7942 (**c**) cultures under white or green light. The growth of the culture is shown as the increase in the optical density of cells at 750 nm (OD$_{750}$, *G. violaceus*) or 730 nm (OD$_{730}$, *S. elongatus*). Cells were illuminated with 10 μmol m$^{-2}$ s$^{-1}$ (*G. violaceus*) or 40 μmol m$^{-2}$ s$^{-1}$ (*S. elongatus*) green light (green line) or white light (black line) LEDs. Values are represented as mean ± s.d. with raw data from three independent experiments. **d**, Low-temperature fluorescence excitation spectra of wild-type *S. elongatus* (black dashed line) and the transformant mild-expressing or overexpressing *pebAB* (PebAB-MX (red dashed line) or PebAB-OX (red solid line)). Fluorescence emission was monitored at 685 nm (mostly PSII fluorescence). The spectra were normalized at their emission peaks. **e**, Competition between PebAB-MX and wild-type cells under green light (green) or white light (black), plotted as a function of the estimated number of generations. The results for seven (green light) or four (white light) independent experiments are shown for each treatment as different lines and symbols. **f**, Fraction of brown colonies that yield PEB to all colonies of PebAB-OX under white and green light. Values are represented as mean ± s.d. from four (white light) or three (green light) independent experiments (closed circles) and raw data (open circles).

We performed numerical simulations based on diffusion equations to predict the distributions of oxygen, reduced iron and iron hydroxide in Archaean aquatic environments. On the basis of the concentrations of reduced iron and oxygen defined by the previous studies[17,24,25], we found an average iron hydroxide concentration of ~10 μM in the oxidized zone (Fig. 1b). On the basis of the previous studies[17,26], we assumed that the oxic and anoxic zones were divided by the pycnocline. Since the concentration of oxygen remained constant as a result of the high eddy diffusivity above the pycnocline, iron hydroxide was probably formed at the boundary between the oxic and anoxic zones and may have continuously spread across the cyanobacterial habitats. Importantly, the distributions of oxygen and reduced iron, crucial for maintaining the underwater light environment, were in equilibrium. The equilibrium concentration mainly depends on the influx of reduced iron to the pycnocline, rather than the conditions in the mixed layer, which served as the cyanobacterial habitat. In fact, oxygen concentration above the pycnocline did not influence iron hydroxide level (Extended Data Fig. 1e,f).

Our experiments on the iron hydroxide formation suggest that iron hydroxide particles were probably small (<100 nm) at the time of formation, allowing them to remain buoyant (Extended Data Fig. 2). As a result, iron hydroxide would have consistently influenced the underwater light environment of the cyanobacterial habitat, while reduced iron was continuously oxidized through photoferrotrophic and cyanobacterial activities, although uncertainty in reduced iron influx could lead to iron hydroxide concentrations varying from 1 to 100 μM (Supplementary Discussion 2).

We also measured the molar absorption coefficient of iron hydroxide, aligning with the previous study[27] and found that its visible-light coefficient remains unaffected by synthesis methods and particle size (Extended Data Fig. 2 and Supplementary Discussion 3). While different types of iron hydroxide might have existed during the Archaean era[28], only the earliest phase of small-sized iron particles probably had the most prolonged impact on the underwater light environment (Supplementary Discussion 5).

Considering that underwater environments could protect cyanobacteria from harmful ultraviolet (UV) radiation in the Archaean era[29], we determined the spectrum of the light window available to photosynthetic organisms. In the absence of iron hydroxide, the cyanobacteria habitat could have extended to depths >20 m. The UV light absorption of iron hydroxide suggests the expansion of cyanobacterial habitats into shallow waters (~5 m), in accordance with the geological records (Extended Data Fig. 2)[30,31]. At both 5 and 20 m depth, the light window predominantly ranges between 500 and 600 nm under 10 μM iron hydroxide concentration (Fig. 1c). The underwater light window mostly remains consistent even with tenfold changes in iron hydroxide concentration (Extended Data Fig. 3). Our analyses show a striking correlation between the spectra of the light window and the green-absorbing phycobilin pigment, PEB, under a broad range of conditions (Fig. 1d,e, Extended Data Fig. 4 and Supplementary Table 1). This implies that the phycobilin pigment could be helpful for harvesting the light in the Archaean era, compared to Chl *a* (Supplementary Discussion 4). It is important to note that this implication applies to various cyanobacterial habitats, such as open oceans, coastal areas and freshwater environments. Additionally, because the green-light window spreads

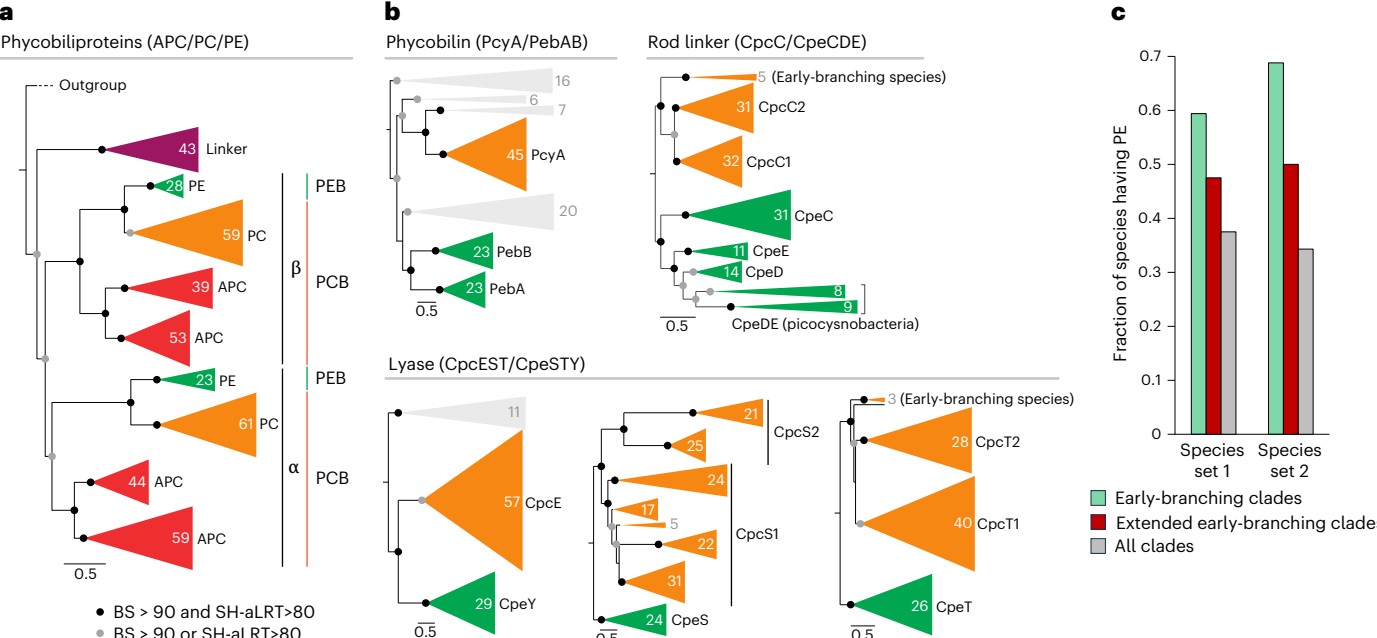

**Fig. 3 | Evolutionary relationship of PC- and PE-associated proteins and change in the relative abundance of PE-bearing species in cyanobacteria.**
**a**, Maximum likelihood phylogeny of phycobiliproteins APC (ApcABDF, red), PC (CpcAB, orange) and PE (CpeAB, green) and the core membrane linker (ApcE, purple). Corresponding phycobilins, PCB and PEB, are also shown. Greek letters indicate the two subunits of phycobiliproteins. The colour code is the same as Fig. 2a, except for ApcE. BS, bootstrap value; SH-aLRT, Shimodaira–Hasegawa approximate likelihood ratio test. **b**, Maximum likelihood phylogeny of other phycobilisome-associated proteins: phycobilin synthase, rod linker (connecting different phycobiliproteins) and lyase (conjugating phycobilins with phycobiliproteins). The subclade collapse in each tree is based on the distribution of corresponding paralogues in host cyanobacteria. Light grey indicates the outgroup or proteins that are not part of phycobilisomes. The green and orange colours represent the absorption wavelengths of PE- and PC-associated proteins, respectively. **c**, Fractions of PE-bearing cyanobacterial species in early-branching clades that branched by the end of the GOE (green), extended early-branching clades that additionally include the taxonomic groups of *Acaryochloris marina* MBIC11017, *Synechococcus* sp. PCC 6312 *and Thermostichus lividus* PCC 6715 (brown) and all extant clades (grey). Calculations were performed for the two distinct species sets, species set 1 (ref. 8) and species set 2 (ref. 9) (Methods).

only a few metres below the surface, both planktonic and benthic cyanobacterial communities would have grown under similar light conditions, assuming that planktonic cyanobacteria were primarily transported by oceanic diffusions, currents and tides. Thus, regardless of the cyanobacterial species and habitats, the co-evolutionary relationship between the light window and aquatic cyanobacteria can be envisioned (Supplementary Discussion 6).

Our findings further include the contemporary light environment around Iwo Island in the Satsuma archipelago[32], where the oxidative conversion of reduced Fe(II) ions emanating from thermal vents created a similar green-light window at a depth of 5.5 m (Extended Data Fig. 5). Intriguingly, at this depth, PEB within cyanobacterial community is more abundant than at the surface (Supplementary Discussion 11), indicating a strong correlation between the light environment and the pigment composition in their natural habitat (Extended Data Fig. 6). This green-light environment was also previously confirmed even in a modern lake with a minor dispersion of iron hydroxide[33].

In conclusion, an underwater green-light environment during the Archaean era was highly plausible, as reduced iron supplied from hydrothermal vents was continuously oxidized at the boundary between oxidized and reduced environments, leading to the formation of BIFs. This specific light condition crucially influenced the evolutionary trajectory of early photosynthetic life.

## Natural selection of phycobilin in green-light environments

Our observations bring to light the synchrony between the green-light window and the absorption spectrum exhibited by PEB-containing phycobiliproteins—PE (Fig. 1c). The light energy absorbed by PE transfers to Chl *a* via the two other phycobiliproteins, PC and APC[5], progressively diminishing as the wavelengths traverse from green (~565 nm) to red (~670 nm) (Fig. 2b). Here, we examine whether PE was a critical component for photosynthesis under the green-light window. We cultivated two cyanobacterial species (see Supplementary Discussion 6 for the selection of species): *Gloeobacter violaceus* PCC 7421, characterized by the co-presence of APC, PC and PE, alongside *Synechococcus elongatus* PCC 7942, endowed only with APC and PC, under two distinct light environments which emulate the green- and white-light windows (Extended Data Fig. 7a). Cell growth patterns reveal an almost indistinguishable proliferation rate for *G. violaceus* under both the green- and white-light conditions (Fig. 2b), contrasting a slower growth rate for *S. elongatus* PCC 7942 under the green-light condition relative to the white-light condition (Fig. 2c)[34]. This observation is consistent with the excitation spectra in cells showing that *G. violaceus* uses green light for photosynthesis more efficiently than *S. elongatus* (Extended Data Fig. 7b).

We further investigated the necessity of the phycobilin pigment PEB that specifically attaches to PE in modern cyanobacteria, by creating genetically engineered *S. elongatus* PCC 7942, capable of producing PEB, but lacking PE. PEB is biosynthesized through two sequential enzymatic steps from biliverdin IXα by *pebA* (15,16-dihydrobiliverdin:ferredoxin oxidoreductase) and *pebB* (phycoerythrobilin:ferredoxin oxidoreductase)[6]. Two transformants mild-expressing and overexpressing the *pebA* and *pebB* genes, denoted as PebAB-MX and PebAB-OX, were green–brown and brown, respectively (Extended Data Fig. 8a,b). For both PebAB-MX and PebAB-OX cells, the excitation spectra (Fig. 2d) revealed an effective channelling of energy absorbed by PEB towards the photosystems via PC and APC, without PE, showing the possibility that PEB attaches to PC as previously suggested[35]. Notably, the

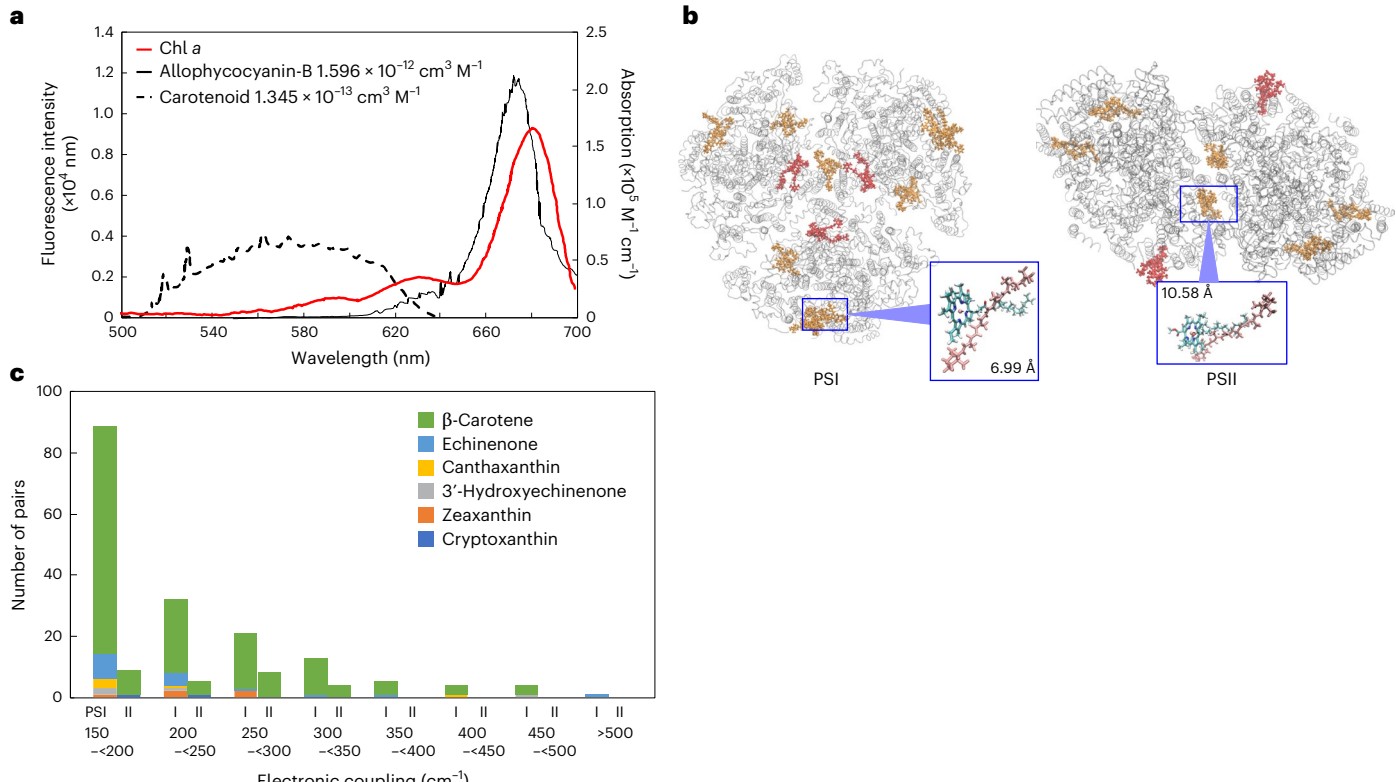

**Fig. 4 | Comparison of excitation energy transfer from carotenoids to chlorophylls in photosystems, as evaluated using quantum chemical analysis.** **a**, The absorption spectrum of Chl *a* (acceptor)[88] and the fluorescence spectra of allophycocyanin-B and carotenoid (donor)[87]. The spectral overlaps between the fluorescence spectra of the donors and the absorption spectrum of the acceptor are shown. **b**, Structure of PSI (PDB code 5oy0) (left) and PSII (PDB code 7n8o) (right) from *Synechocystis* PCC 6803. The pairs of carotenoids and Chl *a* showing from the first-largest to the third-largest electronic coupling are drawn in stick mode in orange (β-carotene) or red (carotenoids other than β-carotene in cyanobacteria). The inset boxes show the zoomed-in views of a pair of β-carotene and Chl *a* exhibiting the largest electronic coupling and the intermolecular distances of these pairs are shown. **c**, Distribution of electronic couplings between Chl *a* and carotenoids in PSI and PSII of cyanobacteria (*Synechocystis* sp. PCC 6803). The number of pairs with electronic couplings >150 cm⁻¹ is shown.

peak intensities that originated from PEB and PC were almost equal in the absorption and excitation spectra of PebAB-MX cells, suggesting well-balanced energy transfer from PEB towards the photosystems (Fig. 2d and Extended Data Fig. 8a). Our observation implies that the use of green light itself does not necessarily require PE, but rather it is sufficient for PEB to simply attach to PC. Yet, the evolution of PEB would have promoted a functional or structural specialization of PC, probably resulting in the evolution of PE and ultimately modern phycobilisomes (Supplementary Discussion 9).

When PebAB-MX cells were cocultured alongside wild-type cells in competition, PebAB-MX cells outcompeted wild-type cells under green light, although the growth under white light was slower in PebAB-MX cells than in wild-type cells because of PCB depletion (Fig. 2e and Extended Data Fig. 8c–e). We also found that some PebAB-OX cells tended to lack brown pigmentation[35] because of spontaneous mutations mainly in the *pebA* gene (Extended Data Fig. 8f). Interestingly, the population of PebAB-OX cells producing PEB was enhanced under green-light conditions, whereas the population of PebAB-OX cells without PEB, because of *pebA* mutations, increased under white-light conditions (Fig. 2f and Extended Data Fig. 8g). These findings substantiate the proposition that, unlike their non-PEB-producing counterparts, cells capable of producing PEB thrived under the selective pressure imposed by the green-light window, reminiscent of natural selection and niche differentiation of photosynthetic organisms caused by differences in pigment composition[10]. While the two species analysed in our study are not representatives of marine cyanobacteria, our arguments are not influenced by the species selection and habitats (Supplementary Discussion 6).

## Phycobiliproteins in ancestral cyanobacteria

Energy transfer from PE towards Chl *a* via PC and APC is key for the efficient photosynthesis under the green-light window. Hence, we hypothesize that PE and PEB were critical for early cyanobacteria to survive under green-light environments before the GOE. Our phylogenetic analyses support the presence of the PE–PC–APC triad already in the common ancestor of crown-group cyanobacteria (Fig. 3a,b and Extended Data Fig. 9; Supplementary Discussion 9). These phycobiliproteins seem to have evolved successively through ancient duplication events of a primordial protein, in the order of APC, PC and PE[36,37]. Unlike the universal distribution of APC and PC in cyanobacteria, PE is sporadically distributed[8,9], due to PE gene loss in many late-diverging species. However, PE exhibits a broader prevalence in early-diverging species (Fig. 3c), suggesting its higher importance during the earlier stages of cyanobacterial evolution. Given the inferred coexistence of all three phycobiliproteins in the common ancestor of modern cyanobacteria, the energy transfer mechanism from PE to the photosystems via PC and APC was probably established within the stem lineage (that is, extinct ancestor lineage) of cyanobacteria under green-light environments before the GOE.

The stepwise evolution of phycobilisomes potentially mirrors a gradual shift in the light window of the Earth from blue to green (see section 'Transition of light window through the history of Earth' below). Ancestral chlorophyll-based photosynthesis probably used the Soret band absorption of Chl *a* spanning the range 350–450 nm (Fig. 1c), but the band cannot be used for green light because of the uphill-type energy transfer. However, green-light energy can still be transferred to Chl *a* via another absorption peak in the red region.

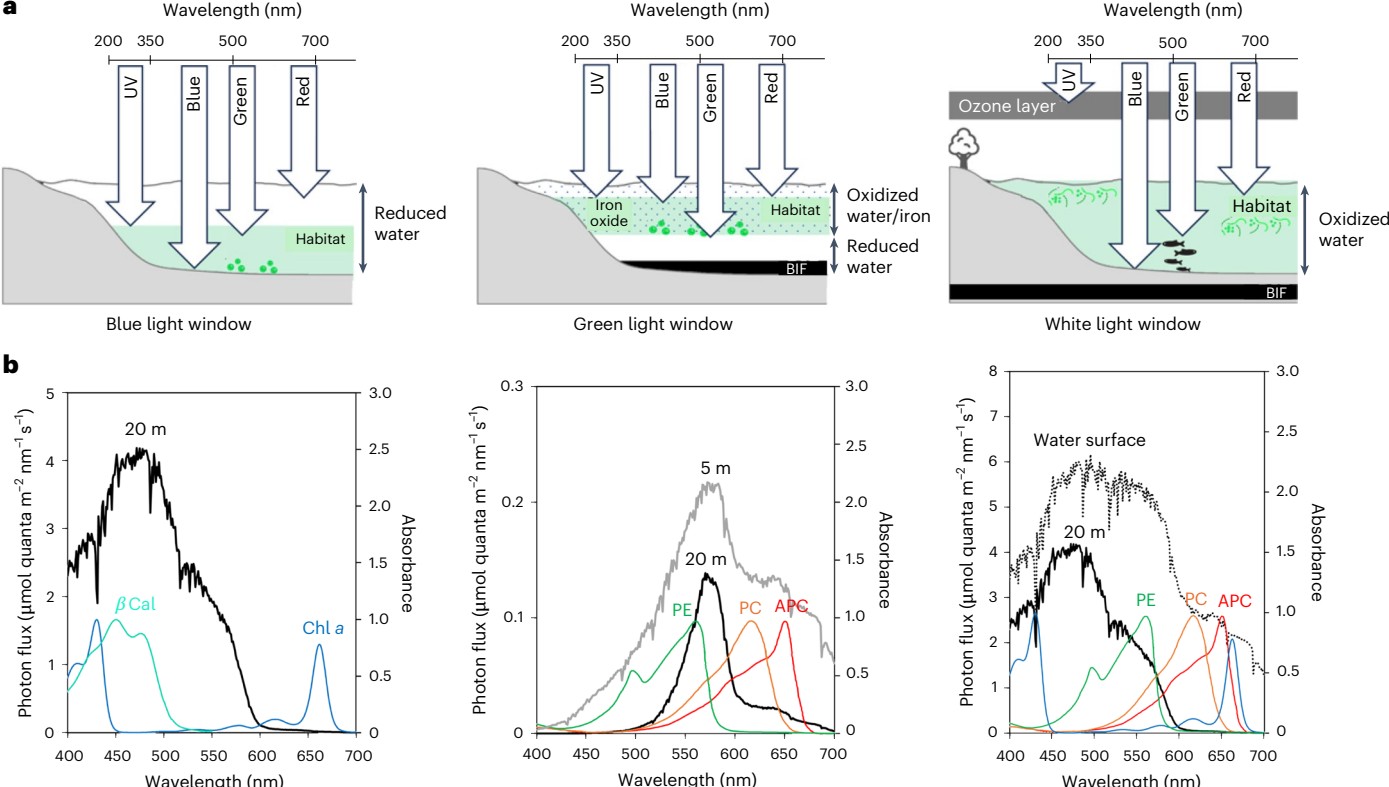

**Fig. 5 | Three light windows for the habitats of photosynthetic organisms.**
**a**, Left, Blue-light window before the emergence of photoferrotrophs and cyanobacteria. Reduced iron was assumed to be spread throughout the ocean. Centre, green-light window due to the formation of iron hydroxide through direct oxidation of reduced iron by photoferrotrophs and indirect oxidation through cyanobacterial-generated oxidized aquatic environment. Iron oxide particles (indicated by polka dots) could efficiently block the UV light, potentially expanding the habitat for photosynthetic organisms into shallow waters. Right, white-light windows after the GOE. The green-shaded region in each panel represents the favoured habitat for photosynthetic organisms from the perspective of blocking the harmful UV light. **b**, Comparison of the transmittance spectrum of the three light windows with the absorption spectra of pigments.

APC and PC have large absorption peaks in the red and orange regions, but they also weakly absorb green light (Fig. 1c). Hence, APC and PC possibly had a selective advantage for host cyanobacteria to use green light, even without PE. The progressive transition to the green-light window would have led to the further adaptation of phycobiliproteins towards PE, which have stronger absorption peaks in the green region than APC and PC, yet their absorption spectra overlap with one another and thus enable the consecutive energy transfer from PE to PC, APC and Chl a.

Depending on the evolutionary timing of iron oxidizing bacteria[38], photoferrotrophy perhaps induced the green-light window before cyanobacterial evolution and thus primitive cyanobacteria were possibly under selective pressure towards green light from early on. Considering that photoferrotrophs can thrive in very low-light environments, it is possible that photoferrotrophs continued to exist in anoxic regions below the pycnocline even after the green-light environment had fully formed (Fig. 1a). At a depth of 50 m under a 10 μM concentration of iron hydroxide, a very faint light environment (~0.1 μmol m$^{-2}$ s$^{-1}$ nm$^{-1}$) might exist, with light levels comparable to those in current green-light environments (~0.01% photosynthetically active radiation (PAR))[33]. Additionally, the weak absorption spectra of bacteriochlorophyll (BChl) a and b, ranging from 550 to 600 nm, fully overlap with the green-light environment.

## Utility of phycobiliproteins for photosynthesis

We further delve into the rationale behind the use of PE and the associated pigment PEB through comparative analyses of PEB with β-carotene, a carotenoid pigment universally distributed in cyanobacteria, in terms of their applicability to photosynthetic antenna systems under green-light conditions. Given that the absorption wavelength peak of β-carotene lies below 500 nm (Fig. 5b, left), PEB is apparently better aligned to absorb green light. The molar extinction coefficient of PEB alone[39] is approximately one-second to one-third that of other pigments such as β-carotene and Chl a (refs. 40,41). However, the covalent attachment of PEB to PE leads to about twofold augmentation of the molar extinction coefficient upon light absorption[39]. Hence, the efficacy of PEB as a light-harvesting antenna is comparable to that of β-carotene, in terms of the absorption coefficients, and PEB is better suited for green-light absorption than β-carotene.

The green-light energy absorbed by PEB in PE is eventually transferred to Chl a in photosystems via another phycobilin pigment, PCB, attached to ApcE or ApcD in APC[42] (Fig. 2a). In this sense, the Förster distance $R_0$, which is the molecular distance that maintains 50% energy transfer efficiency, between PCB and Chl a is another critical factor to determine the overall energy transfer efficiency of phycobilisomes. In our analyses, the Förster distance between PCB in APC and Chl a is found to be approximately seven times longer than that between β-carotene and Chl a (Supplementary Table 2). This is attributed to the high fluorescence quantum yield of PCB and the substantial spectral overlap between APC and Chl a (Fig. 4a and Supplementary Table 2). In general, a larger Förster distance enables energy transfer between more distant molecules. Thus, thanks to the water-soluble phycobiliproteins[5], large light-harvesting antennas such as phycobilisomes can be formed outside host membranes.

Moreover, based on the known core antenna structure of a cyanobacterium *Synechocystis* sp. PCC 6803, the efficiency of energy

transfer from carotenoids, including β-carotene, to Chl $a$ in PSII is found to be much lower than that in PSI (Fig. 4b,c and Supplementary Tables 3–5). These observations suggest that β-carotene is not as effective as phycobilin pigments under the green-light window. Hence, the evolution of phycobilisomes was a reasonable and necessary result of the green-light adaptation of cyanobacterial photosystems.

## Transition of light window through the history of Earth

Our study presents a comprehensive picture elucidating the co-evolutionary trajectory between cyanobacteria and the light environment. Because of the nearly anoxic atmosphere before the GOE[43], the surface of the Earth was potentially exposed to harmful UV-C light >200 nm (ref. 44). Primitive cyanobacteria and photoferrotrophs probably thrived in deeper regions of the photic zone (~20–30 m)[28] because of the lack of protective shields against UV-C radiation[45]. The light window, predominantly influenced by water, favoured a spectrum leaning towards blue, hence termed the 'blue-light window' (left panel of Fig. 5). This could explain the alignment of Chl $a$- and BChl-based photosynthesis with the blue-light window, suggested by the absorption spectrum of Chl $a$ and BChl, matching the spectral range of this window[46,47].

Phycobilisomes probably evolved according to the light environment of cyanobacterial habitats. The stepwise development of phycobilisomes seems to mirror the gradual change in the light window due to the oxidation of the aquatic environment by cyanobacteria or the direct oxidation of reduced iron by photoferrotrophs, which may have initiated before the birth of cyanobacteria[38]. A green-light environment may have additionally been created through the photochemical oxidation of reduced iron (Fe(II)) by UV radiation[48,49]. Resulting Fe(III)-rich minerals might have also protected cyanobacterial habitat from harmful UV radiation (Extended Data Fig. 2). This pivotal transition facilitated the evolution of modern-type cyanobacteria that developed phycobilisomes to transfer the green-light energy to Chl $a$, via three essential phycobiliproteins—PE, PC and APC. Phycobilin pigment biosynthesis may have branched off from chlorophyll biosynthesis, sharing the intermediate protoporphyrin IX, particularly in response to the rise of oxygen, since the intermediate biliverdin IXα is synthesized from protoporphyrin IX via oxygen-dependent haem oxygenase[50] (Supplementary Discussion 9). Capitalizing on the energy endowed by green light, modern-type cyanobacteria may have ascended as primary producers, triggering the GOE.

Following the two oxidation events of Earth, the GOE and the neoproterozoic oxygenation event, the fully oxidized aquatic environment expanded its light window to white light (Fig. 5, right). This transformation resulted from the depletion of oxidized iron species within the aquatic environment and the formation of the ozone layer, thus opening diverse colour niches for phototrophic organisms[10]. Terrestrial cyanobacteria then successfully spread over the entire Earth by adapting to various land conditions[9]. Compared to marine cyanobacteria that thrive under various light environments, including dominant green-light environment, terrestrial cyanobacteria tend to possess only APC and PC[9] because of PE loss as an adaptation to white-light environments on land. Whereas PE becomes dispensable in brighter white-light conditions on land, it retains a more crucial role in dimmer underwater conditions. From the perspective of energy transfer from the light-harvesting antenna to the photosystem, this adaptation suggests that cyanobacteria may have favoured using pigments absorbing light at wavelengths close to those absorbed by the special pair of Chl $a$ in photosystems.

In conclusion, our study traverses through three distinct light windows—blue, green and white—marking different eras in the history of the Earth. This narrative envisions a pre-GOE Earth as a 'pale green dot', symbolizing a potential indicator of aquatic life on distant worlds and contributing to our understanding of the evolutionary journey of photosynthetic organisms.

## Methods

### Calculation of concentration for iron hydroxide

We quantitatively evaluated the concentration of iron hydroxides in cyanobacterial habitats using numerical simulations. The redox state of the ocean has transitioned through the co-evolution of Earth and life, categorized into four epochs[51] (Supplementary Discussion 1). Our study focuses on the second epoch, spanning from the emergence of cyanobacteria to the GOE. During this period, oxygen produced by cyanobacteria oxidized the mixed layer above the pycnocline, aligning with previous findings that oxygen was present in regional areas with higher concentrations, forming 'oxygen oases'[25,52]. Conversely, the layer below the pycnocline remained largely anoxic, as material exchange across this boundary was markedly limited[26,53]. Environments resembling the redox conditions of the Archaean era still exist on present-day Earth, such as Lake La Cruz[54], Lake Paul[55], Lake Matano[17], Lake Pavin[56] and the Red Sea[57]. The distinct boundary layer promotes the production of iron hydroxides in the pycnocline (chemocline), which is largely consistent with the observations in these modern, similar environments[34]. However, some differences exist between the Archaean era and these modern analogues (Supplementary Discussion 12).

Reduced iron from hydrothermal vents was supplied to the pycnocline, where it was oxidized by dissolved oxygen in water[19,58] and by photoferrotrophs[18,20,21]. Iron hydroxide particles, produced through the oxidation of reduced iron, were transported by lateral ocean currents and vertical eddy diffusion before precipitation occurred due to the decoupling of larger particles from the oceanic advection and diffusion processes. While lateral oceanic advection influences the horizontal distribution of iron hydroxide from coastal areas to open oceans, vertical eddies primarily determine the vertical distribution[59,60]. There may have been horizontal gradient in the distribution of iron hydroxide on a global scale. However, the local underwater transmission spectrum is mainly determined by the vertical distribution of iron hydroxide. Therefore, we performed numerical simulations to estimate the vertical distribution of iron hydroxide by considering its formation via chemical reaction, eddy diffusion and precipitation. As reduced iron, primarily sourced from the ocean floor, reaches the boundary between oxic and anoxic zones, iron hydroxide is formed via chemical reactions between oxygen and reduced iron. While reduced iron, oxygen and iron hydroxide are transported solely by vertical diffusion, iron hydroxide also undergoes precipitation. As an initial condition, the calculation box contains oxygen only above the pycnocline. We assumed that the concentrations of oxygen in the oxic region (above the pycnocline) and reduced iron at the bottom of the box are constant over the calculation time. Reduced iron supplied from the bottom of the box spreads throughout the box as a result of vertical diffusion over time. All the parameters used for this calculation were compiled in Supplementary Table 6.

On the basis of the above considerations, we modelled a one-dimensional region with a height of 150 m such that lateral advection is negligible. The following equations were established:

$$\frac{\partial \left[ Fe^{3+} \right]}{\partial t} = \kappa \frac{\partial^2 \left[ Fe^{3+} \right]}{\partial z^2} + k_{ox} \left[ O^2 \right]^{0.58} \left[ OH^- \right]^2 \left[ Fe^{2+} \right] - \frac{2}{3} 10^{-6} \left[ Fe^{3+} \right], \quad (1)$$

$$\frac{\partial \left[ O^2 \right]}{\partial t} = \kappa \frac{\partial^2 \left[ O^2 \right]}{\partial z^2} - k_{ox} \left[ O^2 \right]^{0.58} \left[ OH^- \right]^2 \left[ Fe^{2+} \right], \quad (2)$$

$$\frac{\partial \left[ Fe^{2+} \right]}{\partial t} = \kappa \frac{\partial^2 \left[ Fe^{2+} \right]}{\partial z^2} - k_{ox} \left[ O^2 \right]^{0.58} \left[ OH^- \right]^2 \left[ Fe^{2+} \right], \quad (3)$$

where $[i]$ and $\frac{\partial [i]}{\partial t}$ are the concentration and production rate of the $i$th species, respectively, $\kappa$ is the eddy diffusivity and $k_{ox}$ is the coefficient of the reaction rate between oxygen and reduced iron. The eddy

diffusivity is assigned two different values: a higher value above the pycnocline and a lower value below it (Supplementary Table 7). It is necessary to note that $[Fe^{3+}]$ and $[Fe^{2+}]$ represent the concentrations of iron hydroxide and reduced iron, respectively. The second term of each equation represents iron hydroxide formation under a low oxygen concentration (<10 μM)[61]. The first term of each equation describes the vertical transport of each species through diffusion. The third term of equation (1) represents the removal of iron hydroxide from the calculation area through precipitation based on the particle sedimentation velocity as measured in a previous study[55].

We obtained equilibrium distributions for all three species. Although we assumed that reduced iron is oxidized by the dissolved oxygen in water, the concentration of iron hydroxide remains unchanged regardless of the oxidation process. This is because all the reduced iron can be consumed by photoferrotrophs thanks to its faster oxidation rate than the chemical reaction rate between oxygen and reduced iron (Supplementary Discussion 2).

It is also important to note that silica might have been largely dissolved in the Archaean ocean[21], leading to a much higher concentration of reduced iron even in the oxidized zone. This is because the dissolved silica is transformed into silica–Fe(II) minerals through reaction with reduced iron[62]. As a result, the concentration of iron hydroxide may have been affected by silica. However, because it is difficult to quantitively evaluate the reaction rate between dissolved silica and reduced iron, we did not consider the impact of the dissolved silica on the underwater transmission spectrum, which should be treated by a future study (Supplementary Discussion 5).

## Measurement of iron hydroxide absorption coefficient

Colloidal iron hydroxide was prepared by adding $FeCl_3$ solution to boiling ultrapure water or by adding NaOH solution to $FeSO_4$ solution[27,63]. The final concentration of $FeCl_3$ was 100 μM and $FeSO_4$ solutions were prepared at concentrations of 100 μM or 1 mM, respectively. The solution pH was measured using an Horiba pH meter F-52 (Horiba). The pH values of colloidal suspensions of iron hydroxide prepared from 100 μM $FeCl_3$, 100 μM $FeSO_4$ and 1 mM $FeSO_4$ solution were pH 4, 9 and 7, respectively. At each time after the preparation of iron hydroxide, the transmittance spectra were measured using a V-650 spectrophotometer (Jasco) equipped with an integrating sphere (ISV-722, Jasco), with ultrapure water serving as the reference. It was assumed that all iron formed $Fe(OH)_3$, and the molar absorption coefficient of colloidal suspension of iron hydroxide was calculated from the transmittance as $Fe(OH)_3$.

The particle-size distribution of the aqueous colloidal iron hydroxide solution was evaluated from the autocorrelation function determined using dynamic light scattering (DLS) techniques ($FeCl_3$ and 100 mM $FeSO_4$) or laser diffraction techniques (1 mM $FeSO_4$), respectively. DLS measurements were carried out using an Otsuka ELSZ-2 analyser (Otsuka Electronics). Colloidal suspension of iron hydroxide prepared from $FeCl_3$ at a final concentration of 1 mM was used for DLS measurements, because that of 100 μM was below the detection limit. Laser diffraction measurements were carried out using an Horiba LA-920 (Horiba). Relative refractive index values of 2.39 (iron hydroxide) and 1.33 ($H_2O$ as solvent) were applied for particle-size analysis, according to the instrument manual.

## Calculation of light window

The spectral range of light that permeates the habitat of photosynthetic organisms, referred to as the light window, emerges as a consequence of the interplay between solar irradiance and the transmittance properties of both the atmospheric and aquatic environments. Within the visible spectrum, this light window is primarily governed by the product of the water and atmospheric transmittances. In modern water environments, the absorptions of water, dissolved ion, phytoplankton, non-algal particles and coloured dissolved organic matters primarily govern the underwater transmission spectrum[64]. However, given that

biomass might have been much lower in the Archaean ocean than in modern water environments as a result of very limited availability of phosphorus[65], the absorptions of phytoplankton and coloured dissolved organic matters were ignored. It is also important to note that, although reduced iron was abundant in the Archaean water environments, its absorption coefficient is very low, except for the UV wavelength range[46]. As a result, the effects of only water and iron hydroxide as non-algal particles on the underwater transmission spectrum were considered. On the other hand, there may also exist some other factors affecting the light window. The uncertainties related to the light window analysis are covered in Supplementary Discussion 5.

Regarding the incident solar spectrum at the top of the atmosphere, although there was a 30% increase in the brightness of the Sun over the course of the history of the Earth, the solar spectrum is presumed to have remained largely unchanged[66]. Therefore, using the contemporary solar spectrum in our analysis is unlikely to affect the conclusion of this study. The stable nature of atmospheric transmission across the visible spectrum (400–700 nm) both before and subsequent to the GOE[67] validates the adoption of the air mass 1.5 spectra, as defined by American Society for Testing and Materials.

For the calculation of underwater transmittance, the proportion between the incident irradiance upon the water surface, denoted as $I_0(\lambda)$ and that prevailing at a depth $d$, labelled as $I_d(\lambda)$, is derived through the expression:

$$I_d(\lambda)/I_0(\lambda) = e^{-K_d(\lambda)d}, \qquad (4)$$

where $\lambda$ is the wavelength of the light and $K_d$ denotes diffuse attenuation coefficient. The measurement value of clear ocean water[68] was adopted as the diffuse attenuation coefficient for this calculation. During the era spanning the emergence of cyanobacteria to the GOE, as discussed in the second epoch in Supplementary Discussion 1, the absorption coefficient is predominantly influenced by iron hydroxide in the form of mineral particles and by water itself. The molar absorption coefficient of iron hydroxide, detailed in Supplementary Discussion 3, was thus integrated into our analysis. We incorporated the measured molar absorption coefficient of iron hydroxide into this analysis. On the basis of our numerical simulations, documented in Supplementary Discussion 2, the concentration of iron hydroxide in cyanobacterial habitat varied from 1 to 10 μM for the standard model and possibly increases up to 100 μM. It is important to note that only the absorption coefficient was considered, as the scattering coefficient is deemed constant across the visible spectrum[69].

As discussed in Supplementary Discussion 4, considering that a layer rich in Chl $a$, formed through the balance between light and nutrients, is typically located at the pycnocline and nitracline in open oceans, we set cyanobacterial habitat depths at 50 and 20 m and calculated the incident light spectrum at these same depths (Extended Data Fig. 3a,b). Provided the high molar absorption coefficient of hydroxide in harmful UV light, the habitat could be expanded to the shallow waters (~5 m), consistent with the geological records[30,31] (Extended Data Fig. 3c).

## Sampling site

Water sampling was conducted during cruise of the ship in the sea around Iwo Jima (90 m from shore, 130.320° E, 30.784° N) on 1 November 2023. Water depth, temperature and salinity data were obtained using CTD (smart-ACT, JFE) attached to a Van Dorn water sampler or the spectrometer. Seawater samples were collected for phycoerythrin and Chl $a$ determination, flow cytometry, determination of iron and the measurement of dissolved oxygen (Supplementary Table 8). Transmitted light spectrum was measured with an optical fibre-based compact CCD spectrometer Thorlabs CCS 200 (Thorlabs). On board, dissolved oxygen and pH was measured using Lutron DO-5509 dissolved oxygen meter hydrometer (Mother Tool) and an Horiba LAQUAtwin B-712 pH meter, respectively.

## Determination of iron

Quantification of iron in seawater was performed by conventional phenanthroline colorimetric method. The seawater samples (100 ml) were immediately acidified with hydrochloric acid on board and stored in 100 ml polyethylene bottles. An aliquot of 645 µl of sample or reference solution was mixed with 32 µl of 0.3 M hydroxylamine hydrochloride solution. After the reduction of Fe(III) to Fe(II), 162 µl of sodium acetate solution and 162 µl of 0.25% 1,10-phenanthroline were added. After incubation for 15 min at room temperature, absorbance at 510 nm was measured.

## Flow cytometry

For flow cytometric analyses of cyanobacteria and pigmented nanoflagellates (PNF), water samples were initially fixed with a 2% glutaraldehyde solution, followed by preservation through freezing in liquid nitrogen and subsequent storage at −80 °C for future analysis via flow cytometry[70]. Upon thawing, the frozen samples were subjected to analysis using a Quantum P flow cytometer (Quantum Analysis) equipped with a laser emitting at 488 nm. Samples were processed at a rate of ~5 µl min$^{-1}$ until reaching a volume of 600 µl per sample. The presence of cyanobacteria and PNF was initially verified through fluorescence microscopy. Detection of cyanobacteria and PNF was accomplished by plotting yellow/green fluorescence against orange fluorescence[71]. The positioning of cyanobacteria with distinct pigments on the cytogram was corroborated using wild-type (PC-only) and PEB-MX (PE) S. elongatus PCC 7942 cells, while the placement of PNF on the cytogram was referenced from the previous study[72]. Furthermore, each group was divided into subgroups with high and low fluorescence values based on orange fluorescence.

## Phycoerythrin and Chl *a* determination

One litre of seawater passed through a hand net (150 µm mesh) was filtered onto 47 mm Whatman GF/F filters and kept frozen until analysis. The extraction method of phycourobilin (PUB) and PEB chromophores was modified from the previous study[73]. The filters were cut and extracted in 5 ml of 0.1 mol l$^{-1}$ of NaH$_2$PO$_4$ (pH 6.5) and maintained for 3 h at 4 °C in the dark. Filters were resuspended by vigorous vortex mixing and centrifuged for 10 min at 2,500 rpm. The fluorescence of about 3.5 ml of the supernatant was measured using a Shimadzu RF-5300PC fluorescence spectrophotometer at room temperature (Shimazu). The excitation of phycoerythrin was recorded at 1 nm intervals between 450 and 580 nm (emission fixed at 605 nm). Slit widths were set at 5 nm and 5 nm at excitation and emission, respectively. Excitation spectra of PUB appeared around 495–500 nm while excitation spectra of PEB appeared around 540–560 nm. Chl *a* was extracted in 8 ml of *N*,*N*-dimethylformamide (DMF) by immersing the filter fully into a DMF solution in the dark at −30 °C for 24 h. The samples were measured using a Turner Designs Trilogy fluorometer (Turner Designs) with a CHL-A NA Module (SN: 7200-046-W; excitation at 436 nm, emission at 685 nm).

## Cyanobacterial strains and culture conditions

We used G. violaceus PCC 7421 and S. elongatus PCC 7942 as model species for natural selection experiments. *Gloeobacter* represents the earliest-branching clade (*Gloeobacterales*) within cyanobacteria[14,15,74] and possesses both PE and PC, in addition to APC. It is important to note that *Gloeobacter* only represents Gloeobacteraceae, while Anthocerotibacteraceae is also classified into the order Gloeobacterales[75]. These two clades diverged 1.4 billion years ago and have distinct features: *Anthocerotibacter* lack PE and have paddle-shaped phycobilisomes[76]. By contrast, most cyanobacteria, including the well-studied S. elongatus PCC 7942, possess only APC and PC. The difference in growth conditions between these two taxa are of interest to infer the physiological importance of PE and green light for cyanobacteria. G. violaceus and S. elongatus are extant species and so it remains uncertain whether they

accurately represent ancestral species that might have dominated in the Archaean era. However, because our window analyses apply to any photosynthetic species, regardless of their taxonomic classification and habitats, the model species selection does not affect the main conclusions of our study (Supplementary Discussion 6).

The cultures of G. violaceus PCC 7421 were purchased from the Pasteur culture collection (PCC), while the cultures of S. elongatus PCC 7942 was prepared from our own collection in the laboratory of Nagoya University. G. violaceus PCC 7421 and S. elongatus PCC 7942 were grown in 100 ml conical flasks containing liquid BG-11 medium at 30 °C (ref. 77). For G. violaceus and S. elongatus, cell cultures were grown under static culture conditions and bubbled with air, respectively. For S. elongatus, liquid cultures grown for 3 d after inoculation were used. Cells were illuminated from the bottom of the conical flask with a 5,000 K light-emitting diode (LED) daylight light or green LED light (peak at 520 nm). For G. violaceus and S. elongatus, the light intensity was adjusted to 10 or 40 µmol m$^{-2}$ s$^{-1}$, respectively. It is important to note that the cell cultures stirred by bubbling with air during the experiments had no impact on the transmission spectrum. This observation aligns with the assumption that cyanobacterial abundance might have been low in the Archaean era because of the limited availability of phosphorus[65].

Optical density (OD) was measured as the absorbance of the cell suspension at 750 nm (*G. violaceus*) or 730 nm (*S. elongatus*) using a Genequant100 (GE Healthcare). Both wavelengths measure light scattering by cells without being affected by Chl *a* absorption, so the OD values do not change largely. For the measurement of the OD of G. violaceus, the cell suspension was diluted by half with 40% glycerol to prevent cell sedimentation.

## Spectral analysis

Absorption spectra were acquired using a V-650 spectrophotometer (Jasco) equipped with an integrating sphere (ISV-722). The excitation fluorescence spectra were recorded at 77 K using a FP-6500 fluorescence spectrophotometer (Jasco). For the measurement of G. violaceus, the cell suspension was diluted by half with 40% glycerol to prevent cell sedimentation.

## Construction of PEB biosynthesis mutants

To introduce *pebAB* genes into S. elongatus, *pebAB* genes from a chromatically acclimating cyanobacterium, *Synechococcus* sp. RCC307, were placed under the control of the strong promoter for S. elongatus PC subunit gene clusters, *cpcB1A1* (ref. 78) and introduced into a neutral site II of S. elongatus. For this experiment, the *pebAB* genes were synthesized by Eurofins Genomics according to the sequence reported in the National Center for Biotechnology Information (NCBI) database (locus tag: SynRCC307_2061 for *pebA* and SynRCC307_2062 for *pebB*) and were amplified by polymerase chain reaction (PCR) with primers pebAcpc-f (5′-TTGAAGAATGATGGGATGTTTGATTCCTTCCTCG-3′) and pebB-rv (5′-ACTCTAGAGGATCCGTTACATCCACTTCTTATCAA-3′). The *cpcBA* promoter was amplified by PCR from the genome of S. elongatus, with primers cpcPro-f2 (5′-CTGGCTGGATGATGGGTCGACCATCAACTTAAAG-3′) and cpcPro-rv (5′- CCCATCATTCTTCAAGAAAACTCTCGATTG-3′). The *pebAB* genes and the *cpcBA* promoter were cloned into pNS2KmTΔHincII-Ptrc vector[79]. The DNA fragment including neutral site II and kanamycin-resistance gene were amplified by PCR from the pNS2KmTΔHincII-Ptrc with primers NS2Ptrc-f (5′-CGGATCCTCTAGAGTCGACCTGCAG-3′) and NS2Km-rv (5′- CCATCATCCAGCCAGAAAGTGAGGG-3′). PCR-amplified products of *pebAB* and the *cpcBA* promoter were cloned into PCR-amplified products of pNS2KmTΔ*Hinc*II-P*trc* vector by the In-Fusion HD cloning method (Clontech, Takara Bio). S. elongatus was transformed with the resultant plasmid, named pNS2KmP*cpc*-*pebAB*. PebAB-MX was obtained from PebAB-OX cells with spontaneous mutation showing green–brown colour. The *cpcBA* promoter in PebAB-MX cells had 4 base pair (bp)

(GAAG) insertion 14 bp before the translation start site of *pebA* and had no mutation in *pebAB* genes.

## Competition experiments between PebAB-MX and wild type

Liquid cultures grown for 3 days under white light after inoculation were used. PebAB-MX and wild-type cells were diluted and mixed to an $OD_{730}$ of 0.01. The mixed cultures were grown while being bubbled with air and illuminated from the bottom of the conical flask with an LED daylight light or a green LED light (peak at 520 nm) (40 µmol m$^{-2}$ s$^{-1}$). The cultures were diluted to an $OD_{730}$ of 0.02 every 3–4 or 7 days for white or green light, respectively. To analyse the percentage of PebAB-MX and wild-type cells, small amounts of the PebAB-OX cultures were incubated on solid BG-11 plates with or without kanamycin (10 mg l$^{-1}$) and the numbers of colonies were counted. Only PebAB-MX cells were selected on BG-11 plates with kanamycin, whereas PebAB-MX and wild-type cells appeared on BG-11 plates without kanamycin. The number of generations was estimated from the $OD_{730}$.

## Analysis of spontaneous mutations in PebAB-OX cells

To analyse the spontaneous mutations in PebAB-OX cells, small amounts of PebAB-OX cultures were incubated on solid BG-11 plates and the numbers of green or brown colonies were counted. On 74 randomly chosen colonies, *cpcBA* promoter and *pebA* and *pebB* genes of green colonies of PebAB-OX were sequenced by colony PCR with primers NS2SacIseq-f (5′- CGATAAACGAGCTCGTAAGCGG-3′) and NS2BamHIseq-rv (5′- GCCCTTGCTTTGGGCGATTGAT-3′) using Quick Taq HS DyeMix (Toyobo).

## Phylogenetic analyses

Representative sequences for phycobilisome-associated proteins were identified from UniProt (https://www.uniprot.org/). PE-associated proteins include PebA (Q7NL66), PebB (Q7NL65), CpeA (Q7NLD7), CpeB (Q7NLD6), CpeC (Q7NL63), CpeD (Q7NL62), CpeE (Q7NL61), CpeS (Q7NLD4), CpeT (Q7NLD3) and CpeY (Q7NLD8). PC-associated proteins include PcyA (Q7NHE8), CpcA (Q7M7F7), CpcB (Q7M7C7), CpcC (Q7NM19), CpcS1 (Q7NLD5), CpcS2 (Q7NKE7), CpcT (Q7NLE2) and CpcE (Q7NL58). It is noted that the CpcC homologue in *Anthocerotibacter* is now annotated as CpcJ[80]. These protein sequences were used as enquiries to identify homologous sequences in other cyanobacteria. Sequences were retrieved from GenBank (http://www.ncbi.nlm.nih.gov/) using BLASTp and PSI-BLAST[81,82], with a cutoff threshold of $<1 \times 10^{-5}$. Sequences were aligned using Muscle v.3.8.31 (ref. 83). The amino acid sites that are conserved in <5% of the analysed species were removed, except the carboxy and amino termini, where amino acid sites with <50% conservation were removed. Phylogenetic trees were constructed by maximum likelihood inference using IQ-TREE v.2.1.06 (ref. 84). Substitution models were selected using ModelFinder in IQ-TREE-LG matrix with empirical frequencies estimated from the data (*F*) and the FreeRate model for rate heterogeneity across sites (*R*). Branch support was obtained by Ultrafast bootstrap and SH-aLRT in IQ-TREE.

Species selection was based on a recent phylogenomic study of cyanobacteria[8,9] and taxonomically redundant sequences were excluded. Many species were found to possess multiple homologues for enquiry sequences. Therefore, preliminary phylogenetic analyses were performed and the clustering pattern of those homologues was measured. If individual homologues from a species were separately distributed in different clades, all homologues were retained. By contrast, if those homologues were distributed in the same clades that contain only one species, only the homologue that had the shortest branch length was retained as the representative for the clade. Distantly related proteins that did not constitute phycobilisomes were used as the outgroup in the case of phycobilin synthase (PcyA and PebAB) and lyase (CpcS and CpeS). In the case of CpcS, it was not clear which homologues were involved in phycobilisome formation and hence all clades were taken into account (Fig. 3b).

A few cyanobacteria were found to possess CpeC and CpcC homologues that have a multidomain architecture (about three PBP linker domains) even though CpeC and CpcC generally consist of a single domain. The multidomain architecture was found in, for example, the genera *Leptolyngbya* (CpeC-like; U9W1D6, U9W8A0) and *Gloeobacter* (CpcC-like; Q7NGT2, Q7NL64). Whether these proteins have the same function as CpeC and CpcC is unclear. These multidomain proteins are homologous to each other and are further homologous to ApcE, which generally has a similar multidomain architecture. Hence, the multidomain architecture might represent an ancestral trait of the entire PBP linker protein family (CpeC, CpcC, ApcE and so on). However, the presence of multidomain proteins is punctate within cyanobacteria for both CpeC and CpcC. Also, in all cases, single-domain and multidomain proteins coexist in host organisms. Hence, single-domain CpeC and CpcC appear to be the dominant form, whereas multidomain homologues are probably later acquisitions in a few specific lineages. For phylogenetic analyses, individual domains were separately treated as single-domain proteins. Phylogenetic analyses suggested that CpeC- and CpcC-like multidomain proteins cluster together with single-domain CpeC and CpcC proteins, respectively, rather than with ApcE. Hence, multidomain proteins are inferred to have evolved from gene duplications and subsequent fusions of the CpeC and CpcC genes within individual host organisms. Also, CpeY consists of two HEAT repeat domains. Only the domain at the N terminus is homologous to CpcE and other PBS-related proteins. Hence, the domain at the C terminus was removed.

## Fraction of cyanobacteria possessing the PE gene (*cpeA*)

The fraction of cyanobacteria having PE that branched before and during the GOE was estimated by comparing the distribution of the PE gene (*cpeA*; encoding the PE α subunit)[8,9] in the phylum, using a previously published species tree and Bayesian relaxed molecular clock analysis of cyanobacteria[14,15]. First, we grouped the species listed in two distinct trees[8,9] in terms of the similarity of 16S ribosomal RNA[85]. The 16S rRNA gene sequences of cyanobacterial strains were obtained using the BLAST-N search programme in the NCBI database (http://www.ncbi.nlm.nih.gov). If the 16S rRNA gene sequence information was unavailable, the respective species were excluded from our samples. The 16S rRNA gene identity for two adjacent strains in Fig. 3 in ref. 8 and Fig. 5 in ref. 9 was calculated with the BLAST-N search programme or with ClustalW using the web platform https://npsa-prabi.ibcp.fr/cgi-bin/npsa_automat.pl?page=/NPSA/npsa_clustalwan.html. Species having >95.0% similarity to each other were grouped in the same species cluster. By doing so, our calculation remains unaffected by the sampling number of species. If a cluster contained both species that have and lack *cpeA*, the ratio of the species having *cpeA* was calculated as the fraction of the number of the species with *cpeA* to the total number in the cluster. Second, we selected the taxonomic clades that are inferred to have branched before the GOE, based on the two independent molecular clock analyses[14,15]. However, the taxonomic groups of *Acaryochloris marina* MBIC11017, *Synechococcus* sp. PCC 6312 and *Thermostichus lividus* PCC 6715 were found to have branched nearly simultaneously with the GOE and the date uncertainty prevented us from securely classifying them as early branching. Therefore, we prepared two sets of the early-branching clades: one included these groups (extended early branching; brown bar in Fig. 3c), while the other excluded them (early branching; green bar). Finally, we calculated the fraction of cyanobacteria having *cpeA* relative to the total number of species for both early-branching clades and all extant clades.

## Calculation of electronic coupling, orientation factor and spectral overlap

The energy transfer efficiency is proportional to the spectral overlap between the fluorescence spectrum of the donor and the absorption spectrum of the acceptor. Spectral overlap calculations were performed using the PhotochemCAD 3 programme[86], where β-carotene

and APC-B fluorescence spectra[87] and the Chl *a* absorption spectrum[88] were adopted.

The Förster distance $R_0$ is given by the following equation:

$$R_0 = \left( \frac{9 \left( \ln 10 \right) \kappa^2 \varphi_D}{128 \pi^5 N_A n^4} J \right)^{\frac{1}{6}}, \tag{5}$$

where $N_A$ is the Avogadro constant, $\kappa$ is the orientation factor between the transition dipoles of the donor and acceptor molecules, $\varphi_D$ is the fluorescence quantum yield of the donor molecule, $n$ is the refractive index and $J$ is the overlap between the fluorescence spectrum of the donor molecule and the absorption spectrum of the acceptor molecule. As summarized in Supplementary Table 6, we used $6.00 \times 10^{-1}$ and $6.00 \times 10^{-5}$ for the fluorescence quantum yields of phycobilin[42] and β-carotene[89], respectively; used $1.60 \times 10^{-12}$ and $1.35 \times 10^{-13}$ $M^{-1}$ $cm^3$ for the spectral overlap of phycobilin–Chl *a* and β-carotene–Chl *a*, respectively; and assumed that the other parameters in the Förster distance have the same values between phycobilin and β-carotene. The ratio of the Förster distance of phycobilin–Chl *a* to β-carotene–Chl *a*, $R_0^P/R_0^C$, was then calculated to be 7.01.

The energy transfer efficiency is also proportional to the square of the electronic coupling between the donor and the acceptor molecules. The electronic coupling $V$, an intermediate physical quantity representing intermolecular interactions between different electronic states, was calculated using the transition charge from electrostatic potential (TrESP) method[90] (equation (5)), a typical Coulomb interaction:

$$V = \sum_{i \in A} \sum_{j \in B} \frac{q_i q_j}{4 \pi \varepsilon_0 r_{ij}}, \tag{6}$$

where $q_i$ $(q_j)$ is the transition charge of atom $i$ $(j)$ in molecule A (B); $\varepsilon_0$ is the vacuum permittivity; and $r_{ij}$ is the interatomic distance. The atomic coordinates of PSI and PSII for *Synechocystis* PCC 6803 were obtained from experimentally determined structures (PDB entries 5oy0 and 7n8o). The atomic transition charge $q_i$ of each chromophore was determined using time-dependent density functional theory at the ωB97X/6-311 + G(2d,p) (ref. 91) level in the Gaussian16 programme package[92]. The electronic couplings for all carotenoid–chlorophyll pairs in the PSI or PSII were calculated using equation (6).

The orientation factor $\kappa$ was calculated using the following equation[93]:

$$\kappa = \frac{4 \pi \varepsilon_0 R_{AB}^3}{|\mu_A| \, |\mu_B|} V, \tag{7}$$

where $V$ is the electronic coupling determined by equation (6); $R_{AB}$ is the distance between molecules A and B; and $\mu_A$ and $\mu_B$ are the transition dipole moments of molecules A and B, respectively. The transition moment of molecule A, $\mu_A$, was calculated by equation (8) using the transition charge $q_i$ and position $\mathbf{r}_i$ of atom $i$ in molecule A.

$$\mu_A = \sum_{i \in A} q_i \mathbf{r}_i \tag{8}$$

### Reporting summary

Further information on research design is available in the Nature Portfolio Reporting Summary linked to this article.

## Data availability

All the datasets generated during and/or analysed during this study are available via Figshare at https://figshare.com/s/6d060fb699eaa3d0bc5c?file=51040205 (ref. 94).

## Code availability

All the analysis scripts are available via Figshare at https://figshare.com/s/6d060fb699eaa3d0bc5c?file=51040205 (ref. 94).

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

## Acknowledgements

We express our gratitude to S. Watanabe and M. Watanabe for their invaluable insights and discussions on the biological experiments simulating the natural selection of phycobilisomes under various light environments. We are also grateful to C. Azai, R. Narikawa and S. Takaichi for their fruitful discussions on the role of carotenoids in cyanobacteria. Our appreciation goes to T. Kogiso for his valuable insights into the distribution and dynamics of iron hydroxide during the Archaean era. Lastly, we extend our thanks to P. Sanchez-Baracaldo for her valuable contributions regarding the evolution of light-harvesting antennas from a phylogenetic perspective. This work was supported by Institute for Advanced Research, Nagoya University, JSPS KAKENHI grant nos. 24H00870 and 23H01288, and Astrobiology Center of National Institutes of Natural Sciences (NINS) and JST FOREST program, grant no. JPMJFR202W.

## Author contributions

T.M. proposed the idea that the light window for photosynthetic organisms may have co-evolved with the gradual oxidation of the surface environment of the Earth over its life history. T.M., K.I.-M., Y.H. and H.M. developed an entire picture of how the light window might influence the development of light-harvesting antenna for cyanobacteria, considering the geological evolution and the entire evolution of cyanobacterial stem. T.M., Y.I.F., S.T. and C.A. performed numerical simulations on the distribution of iron hydroxide and calculated the three distinct light windows. K.I.-M. designed and performed biological experiments and analysis. Y.H. performed the phylogenetic analysis. K.J.F. and R.T. performed the quantum chemical calculation. T.M., K.I.-M., C.A., Y.Y. and Y.M. carried out field research on Iwo Island in Kyushu, Japan. T.M., K.I.-M. and Y.H. wrote the initial draft. All authors contributed to the final manuscript. T.M. and K.I.-M. contributed equally to this manuscript.

## Competing interests

The authors declare no competing interests.

## Additional information

**Extended data** is available for this paper at https://doi.org/10.1038/s41559-025-02637-3.

**Correspondence and requests for materials** should be addressed to Taro Matsuo.

[1]Department of Physics, Graduate School of Science, Nagoya University, Nagoya, Japan. [2]Institute for Advanced Research, Nagoya University, Nagoya, Japan. [3]GFZ German Research Centre for Geosciences, Potsdam, Germany. [4]Synchrotron Radiation Research Center, Nagoya University, Nagoya, Japan. [5]Graduate School of Human and Environmental Studies, Kyoto University, Kyoto, Japan. [6]Institute of Transformative Bio-Molecules (WPI-ITbM), Nagoya University, Nagoya, Japan. [7]Department of Chemistry, Graduate School of Science, Nagoya University, Nagoya, Japan. [8]Department of Life Sciences, Faculty of Agriculture, Ryukoku University, Shiga, Japan. [9]Institute for Space-Earth Environment Research, Nagoya University, Nagoya, Japan. [10]Graduate School of Bioagricultural Sciences, Nagoya University, Nagoya, Japan. [11]Department of Life Science & Technology, Institute of Science Tokyo, Yokohama, Japan. [12]Earth-Life Science Institute, Institute of Science Tokyo, Tokyo, Japan. [13]Department of Earth Science, Tohoku University, Sendai, Japan. ✉e-mail: matsuo@u.phys.nagoya-u.ac.jp

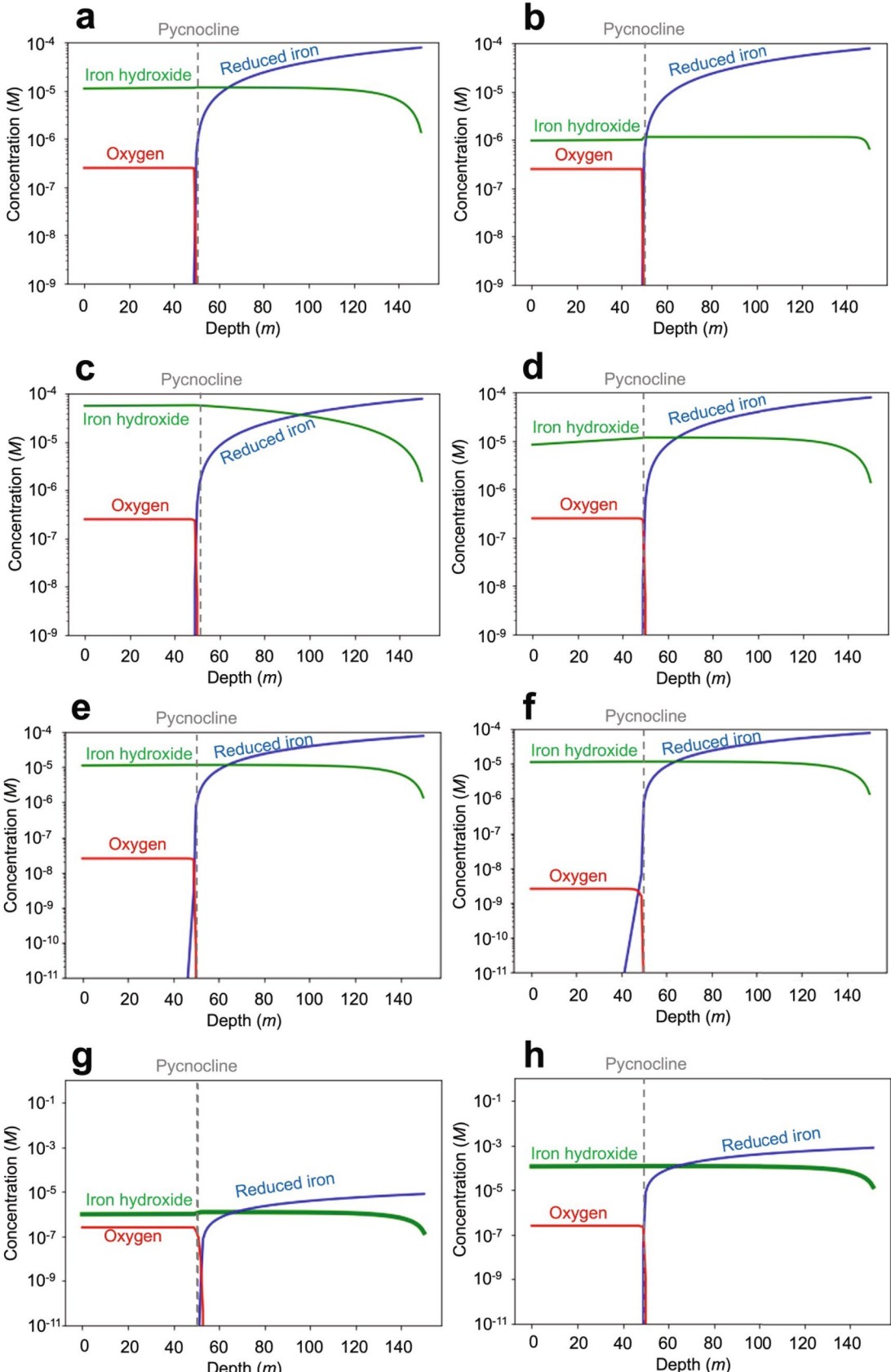

**Extended Data Fig. 1 | Concentrations of oxygen (red), reduced iron (blue), and iron hydroxide (green) in six different scenarios.** This figure illustrates the variations in concentrations across six cases, corresponding to the parameters of Models 1 (**a**), 2 (**b**), 3 (**c**), 4 (**d**), 5 (**e**), 6 (**f**), 7 (**g**), and 8 (**h**) as detailed in Extended Data Tables 1 and 2. Models 1 through 4 are characterized by different diffusivity

in their upper and lower layers, all maintaining a standard oxygen concentration of 250 nM. In contrast, Models 5 – 6 and 7 – 8, while sharing the same diffusivity rates as Model 1, exhibit different oxygen concentrations of 25 and 2.5 nM and different reduced iron concentrations of 8 μM and 800 μM, respectively.

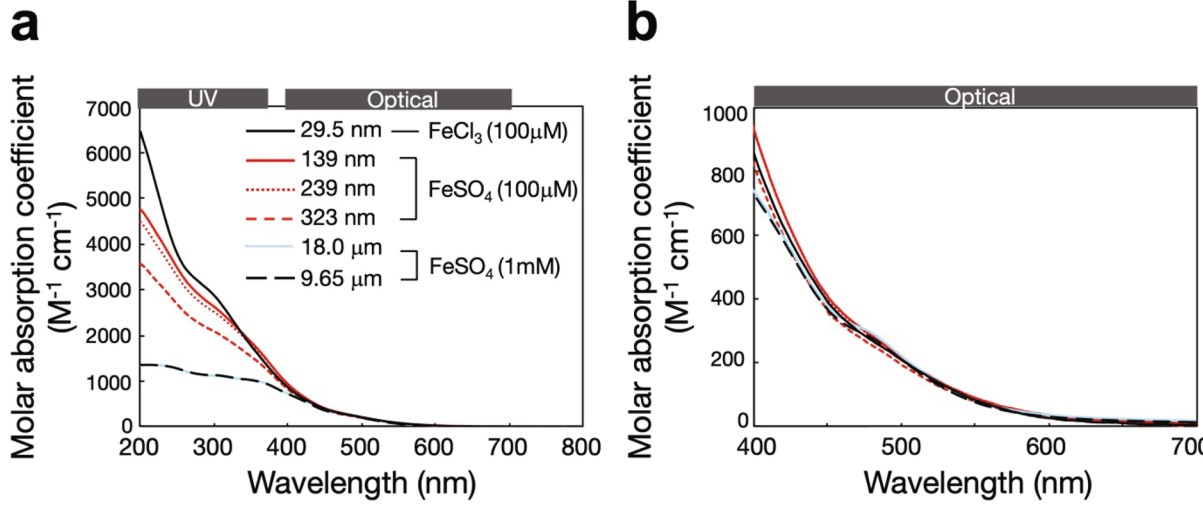

c

| | Reaction time | | Diameter (nm) |
|---|---|---|---|
| FeCl$_3$ (100μM) | | — | 29.5 |
| FeSO$_4$ (100μM) | 0 min | — | 138.7 ± 67.5 |
| | 60 min | ········· | 239.1 ± 141.5 |
| | 120 min | - - - | 322.5 ± 216 |
| FeSO$_4$ (1mM) | 0 min | — | 17960 ± 5550 |
| | 24 hours | — – | 9648 ± 2780 |

All data are expresses as the mean ± SD.

**Extended Data Fig. 2 | Spectra of the molar absorption coefficient determined on samples of iron hydroxide (a) between 200 and 800 nm or (b) between 400 and 700 nm and (c) the diameter of iron hydroxide as a function of the reaction time.** Note that the visible coefficient of iron hydroxide is unaffected by synthesis method and particle size, whereas the coefficient in the UV region increases as particle size decreases. Colloidal iron hydroxide was prepared by adding FeCl$_3$ solution to boiling ultrapure water or by adding NaOH solution to FeSO$_4$ solution. At each time after the preparation of iron hydroxide, the absorption spectra were measured using UV-VIS spectrometer equipped with an integrating sphere. The average particle diameter of iron hydroxide measured by the dynamic light scattering (DLS) method (100 mM FeCl$_3$ and 100 mM FeSO$_4$) or laser diffraction method (1 mM FeSO$_4$) are shown for each reaction time. The absorption spectra and particle size of colloidal iron hydroxide prepared from FeCl$_3$ were measured within 60 min after preparation.

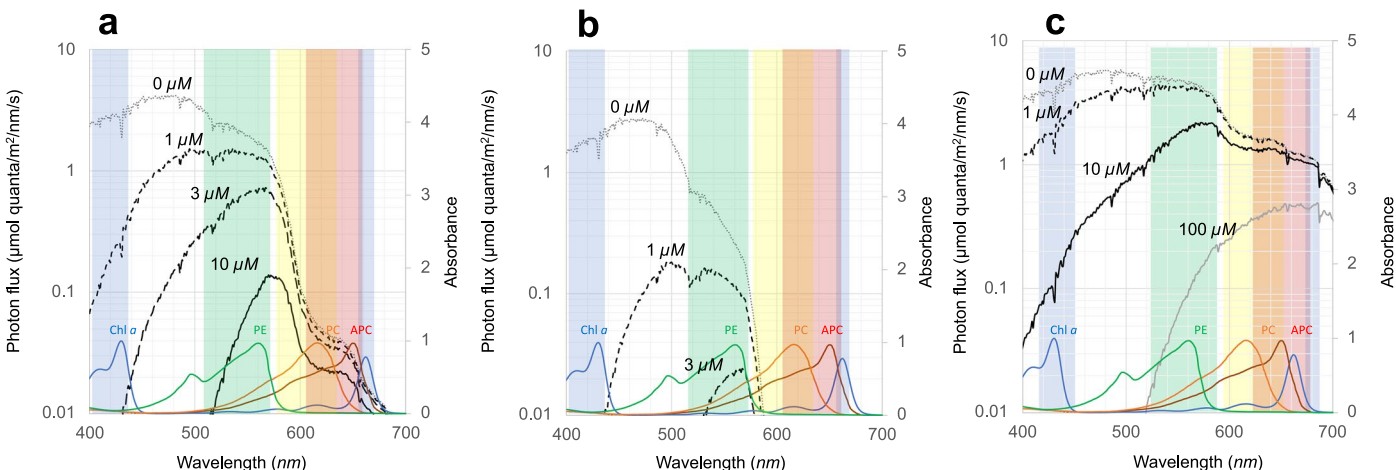

**Extended Data Fig. 3 | Light windows for cyanobacterial habitats at different depths.** This figure depicts the incident photon fluxes in cyanobacterial habitats at depths of 20 meters (**a**), 50 meters (**b**), and 5 meters (**c**). Each habitat's distribution adheres to a normal distribution with a standard deviation (sigma) of 5 meters. Grayscale lines indicate the light windows with different iron hydroxide concentrations: a gray dotted line for the open ocean (0 µM), and black dashed (1 µM), black long dashed (3 µM), solid black (10 µM), and gray lines (100 µM). The cyanobacterial habitat distribution was assumed to be a Gaussian distribution with a half-width of 5 meters. The light window calculations employed a weighted average approach, averaging transmitted photon flux according to the cyanobacterial habitat distribution, which served as the weighting function. Colored regions in the figures denote the absorption wavelength ranges of different pigments: Chl *a* (blue), PE (green), PC (orange), and APC (brown).

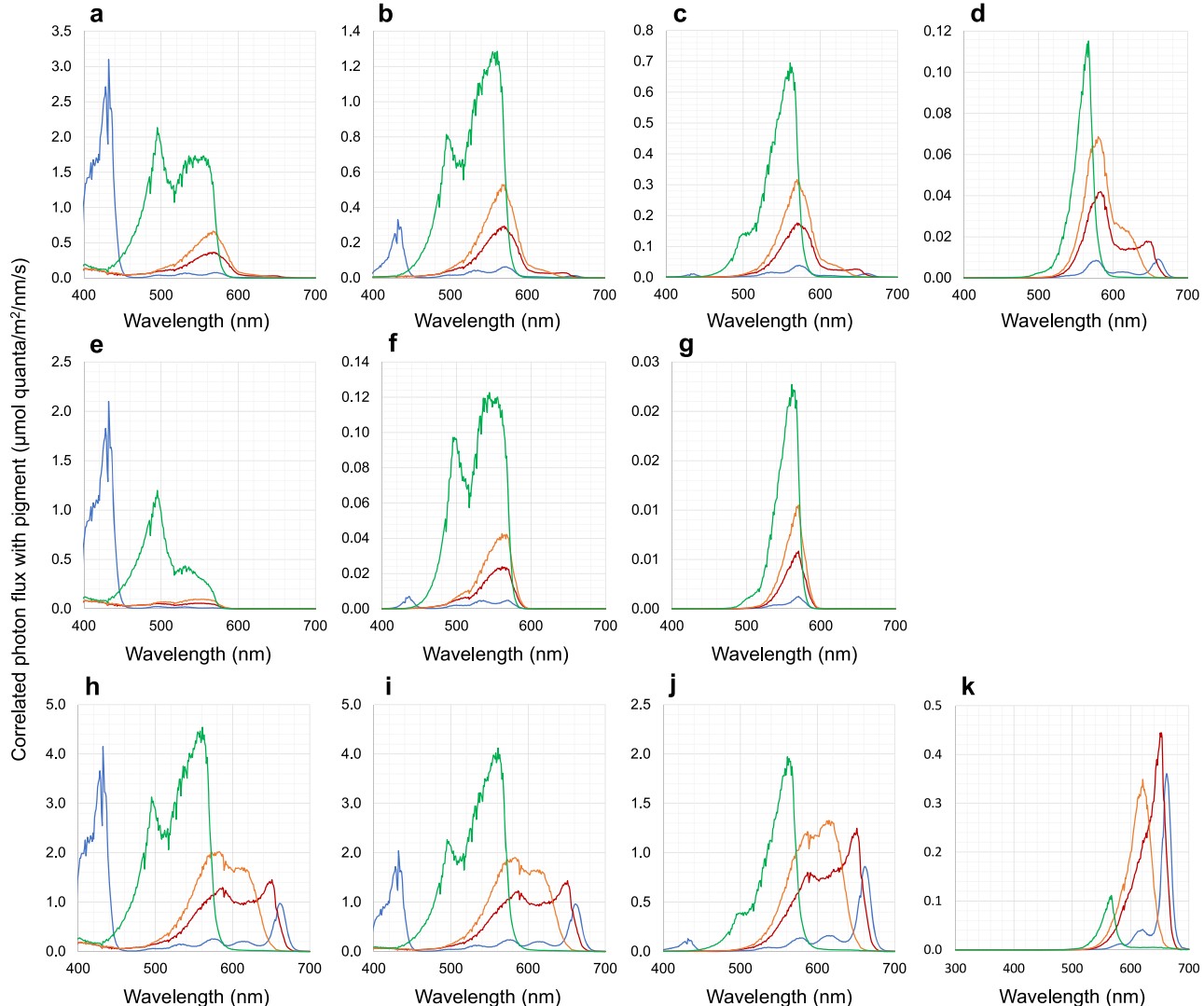

**Extended Data Fig. 4 | Correlation of photon fluxes with photosynthetic pigments at different depths and iron hydroxide concentrations. a–d.** Incident photon fluxes correlated with Chl *a* (blue), PE (green), PC (orange), and APC (red) at a depth of 20 meters. These panels (a: 0 μM, b: 1 μM, c: 3 μM, d: 10 μM iron hydroxide concentration) illustrate the interplay between light window and pigment absorption capabilities under varying iron hydroxide concentrations. The chosen depth of 20 meters aligns with the optimal habitat depth during the Archean era, as indicated by the previous research[29]. **e–g.** Correlated incident photon fluxes at a depth of 50 meters for different iron hydroxide concentrations (e: 0 μM, f: 1 μM, g: 3 μM). This depth represents the typical pycnocline depth in the Pacific Ocean (see Supplementary Discussion 4). **h-k.** Correlated incident photon fluxes at a depth of 5 meters (h: 0 μM, i: 1 μM, j: 10 μM, k: 100 μM), highlighting how varying concentrations of iron hydroxide influence light absorption at this shallower depth.

**a**

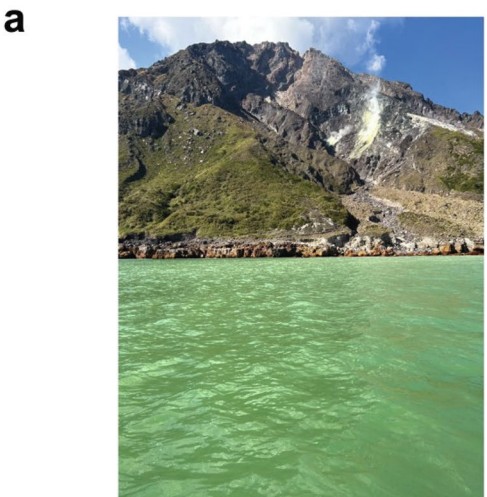

**b**

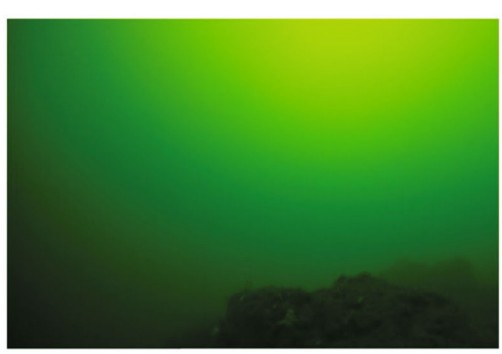

**c**

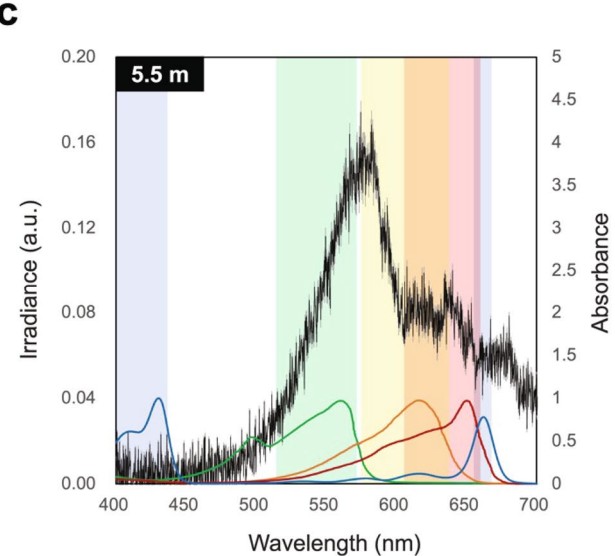

**d**

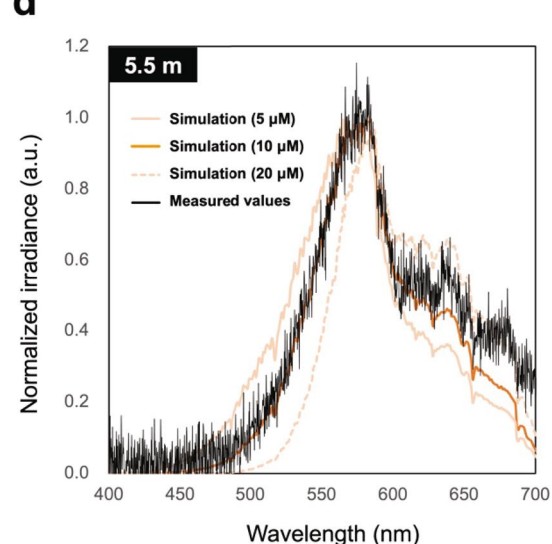

**Extended Data Fig. 5 | Green-light window due to iron hydroxide in the sea around Iwo Jima. a** and **b**. The pictures of sea area (a) and underwater (b) around the Iwo Island within the Satsuma archipelago in Kyusyu, Japan. The underwater picture (b) was provided by the Japanese public broadcaster, Nippon Hoso Kyokai (NHK). **c**. Transmitted light spectrum measured on Iwo Jima at the depth of 5.5 meters. The median value of 10 measurements at the same location is the black line, and the standard deviation is the light gray line. The colored regions in the figures represent the absorption wavelength ranges of various pigments: Chl *a* (blue), PE (green), PC (orange), and APC (red), as in Fig. 3. **d**. Comparison of measured values at Iwo Jima with transmitted light spectrum calculated for an iron hydroxide concentration of 5, 10, and 20 µM. The calculated transmitted light spectrum is set at an equal depth of 5.5 meters. The irradiance just beneath the water surface at Iwo Jima, measured at approximately the same time, was used to calculate the transmitted light. Please note that each spectrum is normalized to the maximum irradiance value of 1 due to the specifications of the measurement device.

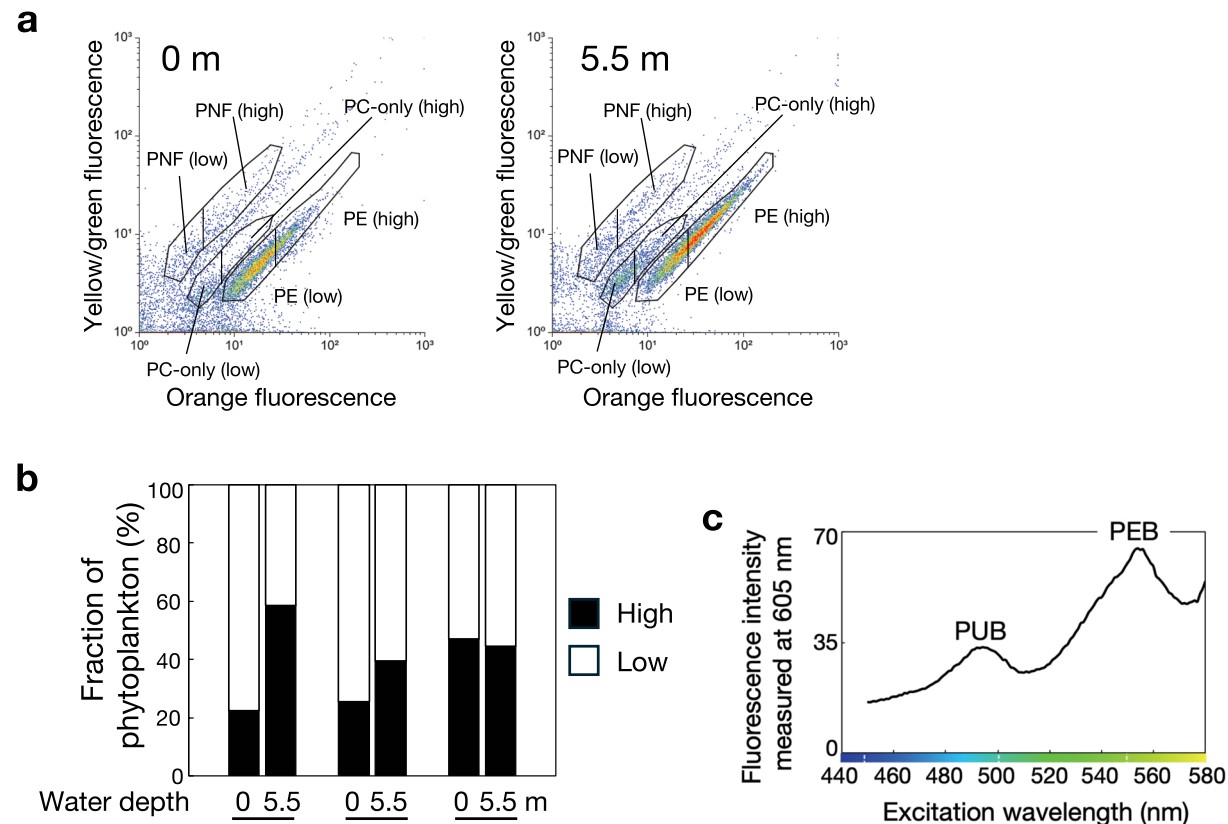

**Extended Data Fig. 6 | Distribution of cyanobacteria in seawater around Iwo Jima. a**. Cytograms of the orange and yellow/green fluorescence profile for samples collected from 0 (left) and 5.5 meters (right) depth at the sea area around Iwo Jima. There were one autotrophic nanoflagellates (ANF) group, cyanobacteria with the PC-only group and cyanobacteria with the PE group. Each group was divided into subpopulations having high or low pigments. **b**. Fraction of phytoplankton having high or low pigments calculated from the subpopulation of cytograms. **c**. Fluorescence excitation spectra of PE in the seawater around Iwo Jima. Two liters of seawater at a depth of 6 meters with a green light environment were filtered onto 47 mm Whatman GF/F filters[70]. The filters were extracted in phosphate buffer, and the fluorescence of supernatant was measured using a spectrofluorometer. Excitation spectra of PUB appeared around 495 – 500 nm and PEB excitation spectra was around 540 – 560 nm. Emission was fixed at 605 nm.

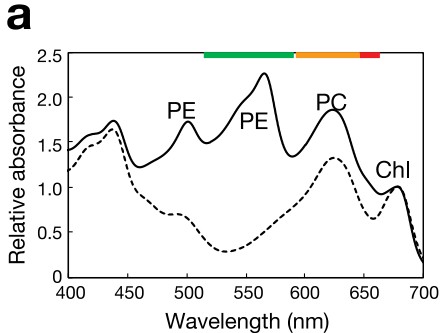

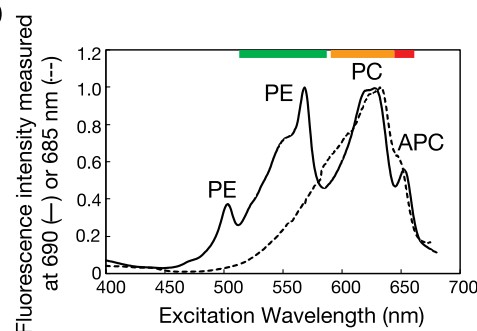

**Extended Data Fig. 7 | Cellular absorption spectra and low-temperature fluorescence excitation spectra of *G. violaceus* PCC 7421 and *S. elongatus* PCC 7942. a.** Cellular absorption spectra of *G. violaceus* PCC 7421 (solid line) and *S. elongatus* PCC 7942 (dashed line). The spectra were normalized at the Chl *a* absorbance at 678 nm. **b.** Low-temperature fluorescence excitation spectra of *G. violaceus* PCC 7421 (solid line) and *S. elongatus* PCC 7942 (dashed line). Fluorescence emission was monitored at 690 nm (*G. violaceus*) and 685 nm (*S. elongatus*) (mostly PSII fluorescence for *S. elongatus* and mostly PSI and PSII fluorescence for *G. violaceus*). The spectra were normalized at their emission peaks.

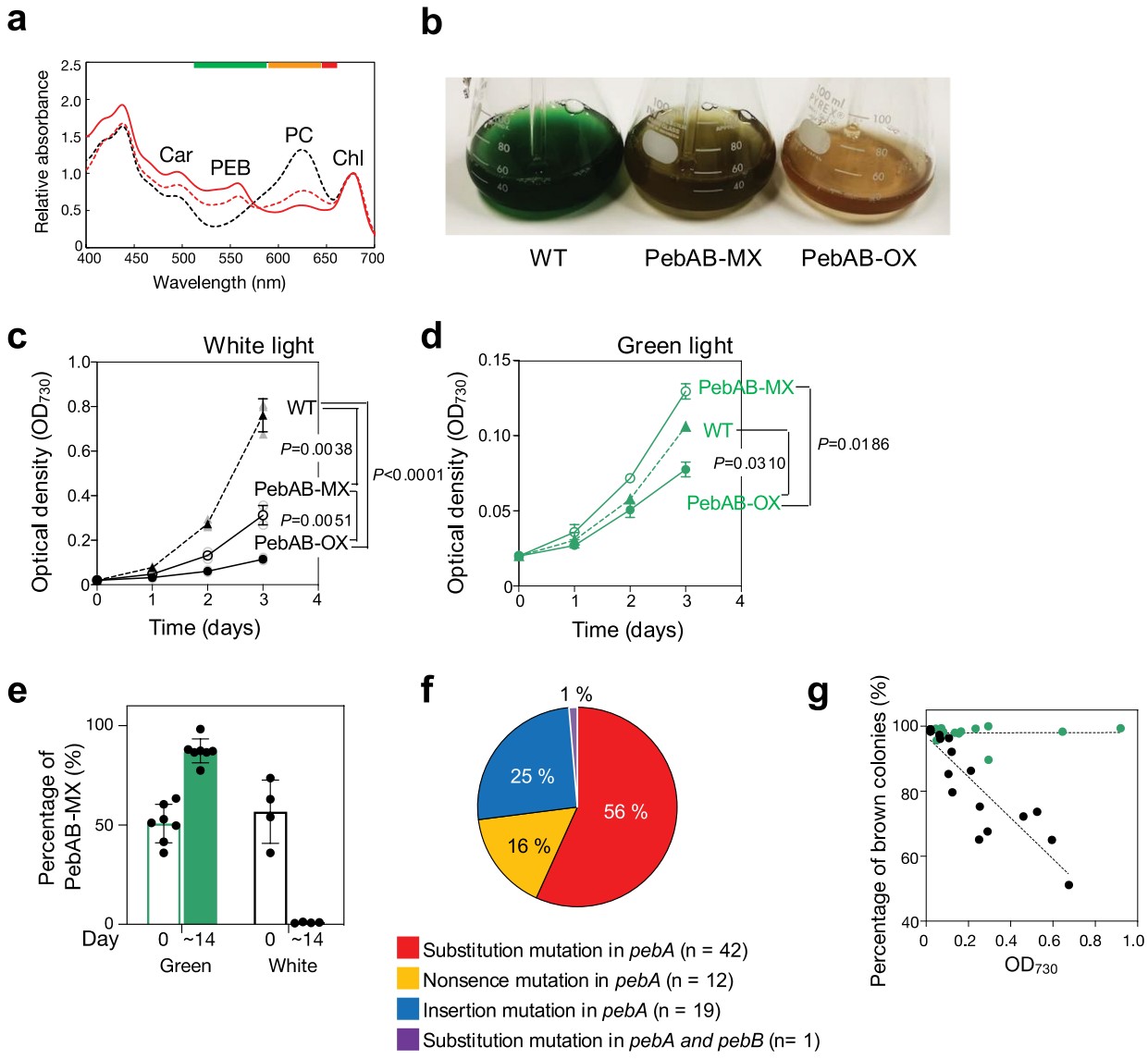

**Extended Data Fig. 8 | Phenotypes and spontaneous mutation of PebAB-MX and PebAB-OX cells. a**. Cellular absorption spectra of wild-type *S. elongatus* (black dashed line), PebAB-MX (red dashed line) and PebAB-OX cultures (red solid line). The spectra were normalized at the Chl *a* absorbance at 678 nm. Abbreviation: Car, carotenoid. **b**. Photograph of the liquid cell cultures of wild-type *S. elongatus*, the PebAB-MX, and the PebAB-OX. **c** and **d**. Growth of wild-type *S. elongatus* (dashed line), the PebAB-MX (solid line with open circles) and the PebAB-OX (solid line with closed circles) under white light (c) and green light (d). Cells were illuminated with 40 μmol m$^{-2}$ s$^{-1}$ green-light or white-light LEDs. Values are represented as means ± SD with raw data from three independent experiments. Statistical significance for growth rate was determined by the one-way ANOVA and the Welch–Brown–Forthyse test, and significant differences are indicated with p values. **e**. Competition between PebAB-MX and wild-type cells under green light (green) or white light (white). Percentages of PebAB-MX cells in the mixture at the start of competition (day 0) or after 14 or 15 days of competition (day 14) are plotted. Values are represented as means ± SD. The results for five (green light) or four (white light) independent experiments are shown for each treatment. **f**. Spontaneous mutations in *pebA* and *pebB* genes in PebAB-OX cells that lack brown pigmentation. Seventy-four green colonies of PebAB-OX cells that lack brown pigmentation were analyzed. **g**. Percentage of brown colonies that yield PEB to all colonies under white light (black) and green light (green), plotted against OD$_{730}$. Results with OD$_{730}$ less than 1.0 are shown.

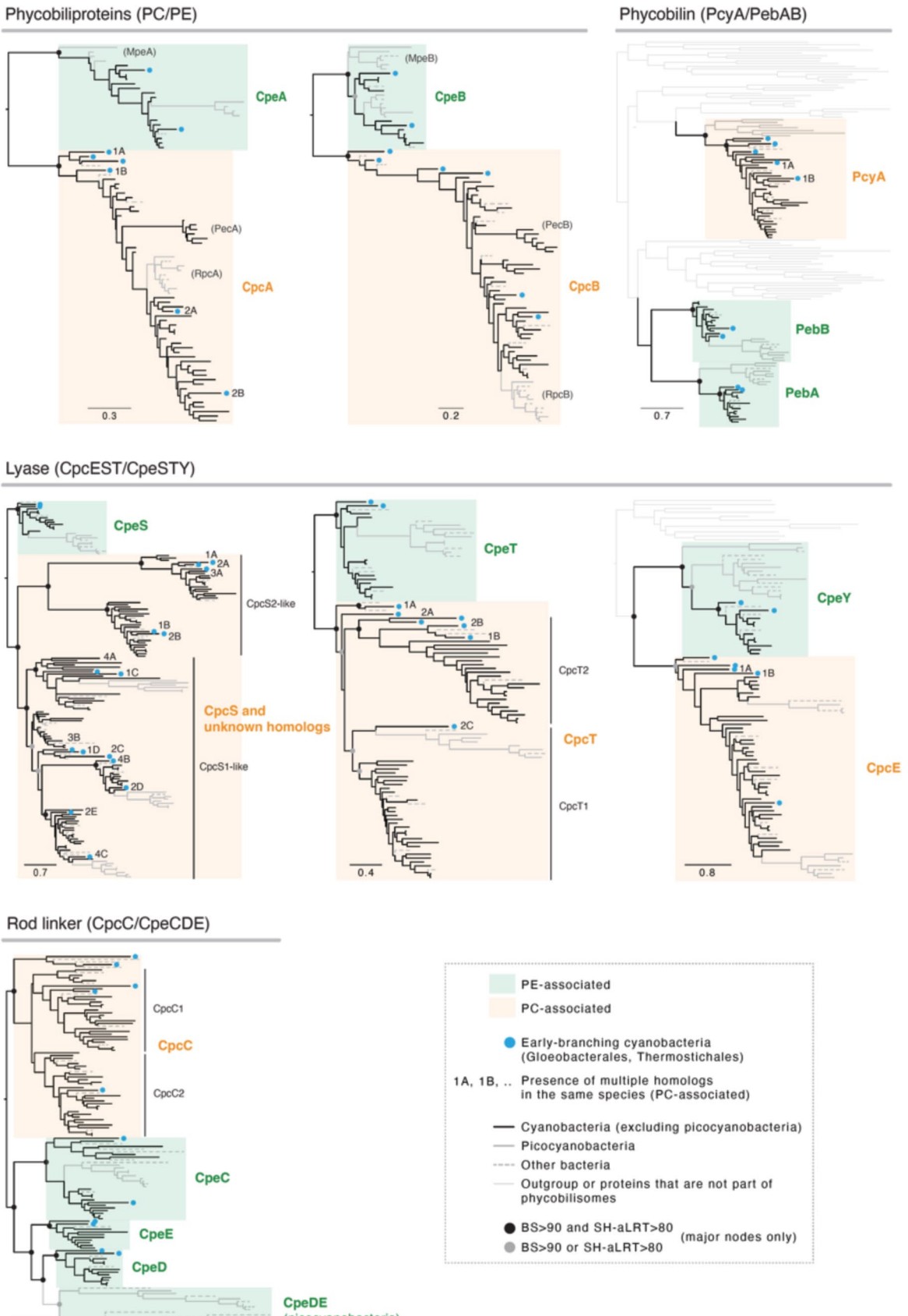

**Extended Data Fig. 9 | Maximum likelihood phylogeny of phycobilisome components.** Phylogenetic analyses include PebAB and PcyA (phycobilin biosynthesis), CpeAB and CpcAB (phycobiliprotein), CpeCDE and CpcC (rod linker), and CpeSTY and CpcEST (lyase).

# Reporting Summary

## Statistics

For all statistical analyses, confirm that the following items are present in the figure legend, table legend, main text, or Methods section.

| n/a | Confirmed | |
|---|---|---|
| ☐ | ☒ | The exact sample size (*n*) for each experimental group/condition, given as a discrete number and unit of measurement |
| ☐ | ☒ | A statement on whether measurements were taken from distinct samples or whether the same sample was measured repeatedly |
| ☐ | ☒ | The statistical test(s) used AND whether they are one- or two-sided<br>*Only common tests should be described solely by name; describe more complex techniques in the Methods section.* |
| ☒ | ☐ | A description of all covariates tested |
| ☐ | ☒ | A description of any assumptions or corrections, such as tests of normality and adjustment for multiple comparisons |
| ☐ | ☒ | A full description of the statistical parameters including central tendency (e.g. means) or other basic estimates (e.g. regression coefficient) AND variation (e.g. standard deviation) or associated estimates of uncertainty (e.g. confidence intervals) |
| ☐ | ☒ | For null hypothesis testing, the test statistic (e.g. *F*, *t*, *r*) with confidence intervals, effect sizes, degrees of freedom and *P* value noted<br>*Give P values as exact values whenever suitable.* |
| ☒ | ☐ | For Bayesian analysis, information on the choice of priors and Markov chain Monte Carlo settings |
| ☒ | ☐ | For hierarchical and complex designs, identification of the appropriate level for tests and full reporting of outcomes |
| ☒ | ☐ | Estimates of effect sizes (e.g. Cohen's *d*, Pearson's *r*), indicating how they were calculated |

*Our web collection on statistics for biologists contains articles on many of the points above.*

## Software and code

Policy information about availability of computer code

| | |
|---|---|
| Data collection | Spectra Manager (JASCO), RFPC (Shimadzu), J-WISE (JFE), ThorSpectra (Thorlabs) |
| Data analysis | Microsoft Excel for Mac v16.77.1, GraphPad Prism 10.0.3, DNA Dynamo v1.616, Muscle v3.8.31, IQ-TREE v2.1.06, PhotochemCAD 3, Gaussian16, Python3 |

For manuscripts utilizing custom algorithms or software that are central to the research but not yet described in published literature, software must be made available to editors and reviewers. We strongly encourage code deposition in a community repository (e.g. GitHub). See the Nature Portfolio guidelines for submitting code & software for further information.

## Data

Policy information about availability of data

All manuscripts must include a data availability statement. This statement should provide the following information, where applicable:

- Accession codes, unique identifiers, or web links for publicly available datasets
- A description of any restrictions on data availability
- For clinical datasets or third party data, please ensure that the statement adheres to our policy

No restrictions apply and all data is available in the manuscript or the supplementary materials. We are now preparing to provide all numerical data, and source file for phylogenic analysis in public repositories (Figshare). And all data are available from the corresponding author upon request.

## Research involving human participants, their data, or biological material

Policy information about studies with human participants or human data. See also policy information about sex, gender (identity/presentation), and sexual orientation and race, ethnicity and racism.

| | |
|---|---|
| Reporting on sex and gender | This study does not involve Human Research participants. |
| Reporting on race, ethnicity, or other socially relevant groupings | This study does not involve Human Research participants. |
| Population characteristics | This study does not involve Human Research participants. |
| Recruitment | This study does not involve Human Research participants. |
| Ethics oversight | This study does not involve Human Research participants. |

Note that full information on the approval of the study protocol must also be provided in the manuscript.

# Field-specific reporting

Please select the one below that is the best fit for your research. If you are not sure, read the appropriate sections before making your selection.

☒ Life sciences   ☐ Behavioural & social sciences   ☐ Ecological, evolutionary & environmental sciences

For a reference copy of the document with all sections, see [nature.com/documents/nr-reporting-summary-flat.pdf](http://nature.com/documents/nr-reporting-summary-flat.pdf)

# Life sciences study design

All studies must disclose on these points even when the disclosure is negative.

| | |
|---|---|
| Sample size | Sample size, number of replicates, error bars and statical tests were chosen based on accepted practices in the field, which are stated in figure legends. All biological experiments were performed independently and reproduced with at least three replicates. |
| Data exclusions | No data was excluded. |
| Replication | All biological experiments was repeated at least three times. Details are provided in Figure legends. |
| Randomization | Samples were not randomized as it was not applicable for the design of this study. |
| Blinding | Investigators were not blinded as it was not applicable for the design of this study. |

# Reporting for specific materials, systems and methods

We require information from authors about some types of materials, experimental systems and methods used in many studies. Here, indicate whether each material, system or method listed is relevant to your study. If you are not sure if a list item applies to your research, read the appropriate section before selecting a response.

### Materials & experimental systems

| n/a | Involved in the study |
|---|---|
| ☒ ☐ | Antibodies |
| ☒ ☐ | Eukaryotic cell lines |
| ☒ ☐ | Palaeontology and archaeology |
| ☒ ☐ | Animals and other organisms |
| ☒ ☐ | Clinical data |
| ☒ ☐ | Dual use research of concern |
| ☒ ☐ | Plants |

### Methods

| n/a | Involved in the study |
|---|---|
| ☒ ☐ | ChIP-seq |
| ☒ ☐ | Flow cytometry |
| ☒ ☐ | MRI-based neuroimaging |

