## [Peer Review File · Nature Ecology & Evolution]

Archean green light environments drove the evolution of cyanobacteria's light-harvesting system

Corresponding Author: Dr Taro Matsuo

A version of this paper was originally rejected for publication by Nature Ecology & Evolution, however that decision was reconsidered after appeal by the authors.

Version 0:

Decision Letter:

20th October 2023

Dear Dr Matsuo

Thank you very much for your enquiry about submitting a manuscript to Nature Ecology & Evolution.

I've now had a chance to discuss your work with my colleagues, and although we think that it sounds very interesting, we are still uncertain as to the degree to which the study will be a good fit for the journal.

Therefore, we would like to invite you to submit the full manuscript to Nature Ecology & Evolution so that we can examine the data before deciding whether to send the paper out to review.

If this is acceptable to you, you can submit the complete manuscript using the link below:

Link Redacted

If you have any questions, please feel free to contact me.

[redacted]

Version 1:

Decision Letter:

31st October 2023

Dear Dr Matsuo

Thank you for submitting your Article entitled "Cyanobacteria Evolution under Aquatic Green World" for consideration. I regret to inform you that after careful consideration and discussion with my editorial colleagues, we have decided that we cannot consider it for publication in Nature Ecology & Evolution.

As you may know, we decline a substantial proportion of manuscripts without sending them to reviewers, so that they may be sent elsewhere without delay. In such cases, even if reviewers were to certify the manuscript as technically correct, we do not believe that it represents a development of sufficient importance to warrant publication in Nature Ecology & Evolution. These editorial judgements are based on such considerations as the degree of advance provided, the breadth of potential interest to researchers and timeliness.

In this case, we do not feel that your paper has matched our criteria for further consideration. We therefore feel that the paper would find a more suitable outlet in another journal. I realise this will be disappointing given our positive response to your presubmission enquiry, but unfortunately we can only make firm decisions based on full manuscripts.

Please be assured that this editorial decision does not represent a criticism of the quality of your work, nor are we questioning its value to others working in this area. We hope that you will rapidly receive a more favourable response elsewhere.

You might want to consider our sister journal [Nature Communications](https://www.nature.com/ncomms/about) as a potential venue for the publication of these results. *Nature Communications* publishes high quality and influential research and across the full spectrum of the natural sciences. More information on the journal, the potential benefits of transfer and a link to transfer your paper, can be found at the bottom of this email. Please note that the editorial team at *Nature Communications* will consider your manuscript independently of our suggestion to transfer.

I am sorry that we cannot respond more positively on this occasion.

[redacted]

*****Nature Communications* is the Nature Research flagship Open Access journal. The editors at *Nature Communications* will assess your manuscript's suitability for potential publication; they aim to provide feedback quickly, with a median decision time of 8 days for first editorial decisions on suitability. The journal is also proud to offer double blind and transparent peer review options. Our [open access pages](http://www.nature.com/ncomms/open_access/index.html) contain information about article processing charges, open access funding, and advice and support from Springer Nature.**

****Although we cannot offer to publish your manuscript, I suggest that you consider Nature Communications as a suitable venue for this work. To transfer your manuscript, please use our manuscript transfer portal. You will not have to re-supply manuscript metadata and files, unless you wish to make modifications. For more information, please see our [manuscript transfer FAQ](http://www.nature.com/authors/author_resources/transfer_manuscripts.html?WT.mc_id=EMI_NPG_1511_AUTHORTRANSF&WT.ec_id=AUTHOR) page.**

****For Nature Research general information and news for authors, see <http://npg.nature.com/authors>.**

Version 4:

Decision Letter:

30th July 2024

Dear Dr Matsuo,

First, I'd like to apologise again for the delay in getting this decision to you. Your manuscript entitled "Evolution of Cyanobacteria under Archean Aquatic Green World" has now been seen by three reviewers, whose comments are attached. The reviewers have raised a number of concerns which will need to be addressed before we can offer publication in *Nature Ecology & Evolution*. We will therefore need to see your responses to the criticisms raised and to some editorial concerns, along with a revised manuscript, before we can reach a final decision regarding publication.

We therefore invite you to revise your manuscript taking into account all reviewer and editor comments. Please highlight all changes in the manuscript text file.

* If you have not done so already please begin to revise your manuscript so that it conforms to our Article format instructions at <http://www.nature.com/natecolevol/info/final-submission>. Refer also to any guidelines provided in this letter.

Link Redacted

Nature Ecology & Evolution is committed to improving transparency in authorship. As part of our efforts in this direction, we are now requesting that all authors identified as 'corresponding author' on published papers create and link their Open Researcher and Contributor Identifier (ORCID) with their account on the Manuscript Tracking System (MTS), prior to acceptance. ORCID helps the scientific community achieve unambiguous attribution of all scholarly contributions. You can create and link your ORCID from the home page of the MTS by clicking on 'Modify my Springer Nature account'. For more information please visit www.springernature.com/orcid.

[redacted]

Reviewer expertise:

Reviewer #1: geomicrobiology, evolution of the earth system

Reviewer #2: Evolution of photosynthesis in cyanobacteria

Reviewer #3: cyanobacterial phylogenetics.

Reviewers' comments:

Reviewer #1 (Remarks to the Author):

Please see attached

Reviewer #2 (Remarks to the Author):

The authors provide an exciting work suggesting the reason for the occurrence of phycobilisome in the evolution of cyanobacteria. The authors used multidisciplinary approaches from geology, phylogeny, and experimental biology to draw their conclusions, which is very impressive. The Supplementary Discussions are also helpful for readers to understand many of the questions left in the main text. The authors' findings will interest people working on photosynthesis, cyanobacteria, or evolution. The study provides an intriguing explanation of the occurrence of phycobilisome, and its uniqueness for adaptation in the Archean Ocean may be why phycobilisome is absent in land plants.

The manuscript is worth publishing. Here are a few suggestions for the authors to address before publication.

1. Terrestrial cyanobacteria can be added to the discussion. Several cyanobacteria in the phylogeny analyses (Source data for Figure 1a, 3b, and Extended Data Figure 9) are terrestrial cyanobacteria, such as *Gloeobacter*, *Anthocerotibacter*, *Leptolyngbya*, *Chroococcidiopsis*, and *Nostoc*. The authors briefly discuss this in Supplementary Discussion 6. However, it would be nice to include the idea in Figure 4 that terrestrial cyanobacteria thrive in the white light spectra on land.

2. The authors can discuss the widespread of *Prochlorococcus* in the modern ocean. This strain has divinyl Chl a and Chl b and no phycobilisome, which differs from other crown cyanobacteria in pigment composition. The different pigment composition might be an adaptation to the modern ocean's white light window.

3. Line 501 and 511: "a" in chlorophyll a should be italicized.

3. Line 517: Cyanobacterial strains and culture conditions. Please describe the source of these two cyanobacterial strains (purchased from a culture collection center or gifted by other labs?)

4. Line 518: Based on the recent classification, *Gloeobacteraceae* and *Anthocerotibacteraceae* were classified into the order *Gloeobacterales*. In the earliest-branching clade (*Gloeobacterales*), *Gloeobacter* only represents the *Gloeobacteraceae* but not *Anthocerotibacteraceae* because these two clades diverged 1.4 billion years ago and have distinct features (*Anthocerotibacter* does not have PE and have paddle-shaped phycobilisomes).

These two references can be added in this paragraph.

Strunecký O, Ivanova AP, Mareš J (2023) An updated classification of cyanobacterial orders and families based on phylogenomic and polyphasic analysis. *J Phycol* 59 (1):12-51.

Rahmatpour N, Hauser DA, Nelson JM, Chen PY, Villarreal A JC, Ho M-Y, Li F-W (2021) A novel thylakoid-less isolate fills a billion-year gap in the evolution of Cyanobacteria. *Curr Biol* 31 (13):2857–2867.e2854.

Gloeobacter violaceus PCC 7421 was isolated from calcareous rock in Switzerland, not from the ocean. As the authors' model of cyanobacteria evolution is in the ocean, *Gloeobacter violaceus* PCC 7421 might not be an ideal model organism to represent early-diverging cyanobacteria in the ocean.

5. Line 528: Please justify using two different wavelengths (750 nm and 730 nm) to measure the optical densities of *G. violaceus* and *S. elongatus*.

6. Source data for Figure 3b and Extended Data Figure 9:

Anthocerotibacter panamensis does not have CpcC. The CpcC sequence of *Anthocerotibacter panamensis* used in the phylogeny has an updated annotation as CpcJ based on a recent publication last year. Please refer to the following reference:

Jiang H-W, Wu H-Y, Wang C-H, Yang C-H, Ko J-T, Ho H-C, Tsai M-D, Bryant DA, Li F-W, Ho M-C, Ho M-Y (2023) A structure of the relict phycobilisome from a thylakoid-free cyanobacterium. *Nat Commun* 14 (1):8009.

Reviewer #3 (Remarks to the Author):

The authors described the fascinating evolutionary process of cyanobacteria under the Archean green aquatic environments before and after GOE. The emergence of cyanobacteria led to the oxidation of Fe(II), forming precipitates and contributing to BIF. The precipitation of Fe(III) shaped the aquatic world into a green one. The authors made a hypothesis that these underwater green light environments drove the evolution of cyanobacteria light-harvesting system because cyanobacteria acquired a green-specialized phycobin and made the transition from green light harvesting to red light usage. They made genetic engineering and phylogenetic analysis of all key components of phycobilisomes of extant cyanobacteria to verify this hypothesis.

Major points:

1. Could the simulation and calculation of light wavelength and light windows in different depths reflect the real situation of Archean aquatic world? Different pH, ion composition, atmospheric conditions and biotic environment dynamically controlled the Archean aquatic light condition. Please provide some explanations.
2. Genetic engineering on *Synechocystis* sp. PCC6803 resulted in the green-light adaptation of this no-PE cyanobacteria. But the growth of transgenic cyanobacteria changed the color of water and culture medium. Will the changes influence the results of genetic engineering simulation of PE-contained *Synechocystis* sp. PCC6803?
3. I am very curious that green light occupied a large area of visible spectrum and have higher energy, but why did cyanobacteria and almost all the photosynthetic organisms choose the red light rather than the green light as one of the main energy sources?
4. For the phylogenetic analyses, it is more convincing to add some analyses or discussion for the divergence, duplication and gain-and-loss events of gene families, rather than simple deduction. For example, in line 156: Given the importance of the PE gene, the discussion of how to evaluate the loss of the PE gene in many late-diverging species is missing here.

Minor points:

1. Line 592: There doesn't seem to be enough details provided here to allow for reproducibility (how was the trimming, masking, etc, done?)
2. Line 596: Except the ultrafast bootstrap (UFBOOT), it would be useful to add the branch supports using SH approximate likelihood ratio (SH-aLRT test) approach.
3. Line 602-603: What are the criteria for selecting representative sequences?
4. Line 628-629: why a previously published molecular clock analysis is being used here, this time frame is robust? Perhaps using your dataset to infer divergence time is better.
5. Figure 3 represents the phylogenetic analysis of phycobiliproteins, bilin biosynthetic enzymes and bilin lyase. What information does the Rod linker phylogenetic tree in Figure 3b intend to convey? It does not match the description in lines 606-624. The information given in the figure legend of Figure 3 is limited, making it rather difficult to understand. Please add more specific information.
6. Line 603-605: I do not understand why are the sequences outside the clades that contain known PE and PC proteins excluded, but the outgroup sequences, non-cyanobacterial PcyA/PebAB homologs and unknown CpcS Homologs are exceptions.

*****END*****

Version 5:

Decision Letter:

29th October 2024

Dear Dr. Matsuo,

Thank you for submitting your revised manuscript "Evolution of Cyanobacteria under Archean Aquatic Green World" (NATECOLEVOL-23102497D). It has now been seen again by the original reviewers and their comments are below. The reviewers find that the paper has improved in revision, and therefore we'll be happy in principle to publish it in Nature Ecology & Evolution, pending minor revisions to satisfy the reviewers' final requests and to comply with our editorial and formatting guidelines.

[redacted]

Reviewer #1 (Remarks to the Author):

I find that the authors have made an excellent attempt at addressing my initial comments. As such, I am happy to accept as is, pending a thorough edit of the text itself.

Reviewer #2 (Remarks to the Author):

The reviewer appreciates the chance to evaluate the revised manuscript. The authors have carefully responded to the primary inquiries and suggestions, significantly improving the manuscript. The reviewer is delighted to confirm that this work is an exceptional contribution to the evolution of phycobilisomes and cyanobacteria.

Reviewer #3 (Remarks to the Author):

Dear Authors,

Thank you for this revised version of the manuscript, and your great efforts to address comments from myself and other referees. I appreciate that the authors provided more concrete analysis and discussions for the loss of PE in terrestrial cyanobacteria and its existence in marine cyanobacteria.

I only have several minor comments.

1. Main Text in L750-752: "Distantly related proteins that did not constitute phycobilisomes were used as the outgroup. In the case of phycobilin synthase (PcyA and PebAB) and lyase (CpcS and CpeS)". Please rephrase this sentence.
2. Lines 733-735: How to distinguish whether similar sequences identified by BLAST are homologs instead of different transcripts?
3. Lines 748-750: Please explain the reason for 'only the homolog that had the shortest branch length was retained as the representative for the clade'. Is the aim to remove redundant transcripts? If so, there is also the possibility of discarding duplicated genes in the genome, how could avoid this situation?

Dear Prof. Kurt Konhauser,

Thank you very much for evaluating our study and for providing such insightful and constructive comments. Your feedback has significantly improved the scientific quality of our manuscript, and we are truly honored to receive suggestions from a world-leading expert like you.

We have also received a number of useful comments from the other reviewers. We have carefully considered all the reviewer's suggestions and revised the manuscript. All revisions in both the main text and supplementary information are highlighted in yellow. Before detailing our responses to each comment, we would like to provide an overview of the major revisions made to the main text and supplementary information. A summary of the major revisions is as follows:

1. We have expanded our discussion in both the main text and supplementary information on how the green light environment—induced by the direct oxidation of reduced iron by photoferrotrophs—could influence the coevolutionary relationship between cyanobacteria and their light environment.
2. The assumptions for our numerical calculations on the distribution of iron hydroxide have been extensively clarified in the methods section. Additionally, we have added a panel (Figure 1a) illustrating the water environment during the Archean era.
3. We have incorporated a detailed discussion on the potential adaptation of cyanobacteria to white light environments after the two oxidation events.
4. A new section has been added to the Supplementary Information to address uncertainties in the light window analysis during the Archean era (Supplementary Discussion 5).
5. We have also discussed the influence of cyanobacterial habitat on our main conclusions in another newly added section of the Supplementary Information (Supplementary Discussion 6).
6. Figure 3c, which shows the fraction of species bearing PE as a function of time, has been revised to account for uncertainties in previous molecular clock analyses. The corresponding method section has also been updated.
7. We have entirely updated our phylogenetic analyses, replacing Figure 3, Extended Data Figure 9, and their source files with the new results.
8. Lastly, we have made minor grammatical adjustment throughout the manuscript to improve readability.

We have addressed each comment from you in detail below. Our responses, as well as the newly inserted or modified sentences of the main text and supplementary information, are highlighted in blue. We hope these revisions meet your expectation, and we would be very delighted to discuss future studies related to this study with you.

Once again, we sincerely appreciate your time and efforts in reviewing this study.

Sincerely yours,

---- Response to each comment ----

Major comments:

- It is unclear why the authors focus on diffusion of ferrihydrite versus advection by currents? I have no doubt that ferrihydrite particles would be carried somewhat laterally away from shore via currents, but some certainly was deposited because we get banded iron formation at that time. So, how does diffusion versus advection change Figure 1, if at all?

RESPONSE: Thank you very much for raising this concern. We agree with your point that the distribution of iron hydroxide is influenced not only by vertical eddy diffusion but also by lateral advection. Lateral advection could result in a non-uniform distribution of iron hydroxide concentration, potentially varying with distance from the shore. However, considering that the underwater transmission spectrum is primarily determined by the vertical distribution of iron hydroxide, we focused on a one-dimensional framework to predict it. Using this framework as a basis, we plan to develop a three-dimensional simulation in future studies. Based on these considerations, we have added Figure 1a and have extensively detailed the assumptions made for the calculations in the methods section as follows:

L477 in Main Text:

Reduced iron from hydrothermal vents was supplied to the pycnocline, where it was oxidized by dissolved oxygen in water^{18,58} and by photoferrotrophs¹⁹⁻²¹. Iron hydroxide particles, produced through the oxidation of reduced iron, were transported by lateral ocean currents and vertical eddy diffusion before precipitation occurred due to the decoupling of larger particles from the oceanic advection and diffusion processes. While lateral oceanic advection influences the horizontal distribution of iron hydroxide from coastal areas to open oceans, vertical eddies primarily determine the vertical distribution^{59,60}. There may have been horizontal gradient in the distribution of iron hydroxide on a global scale. However, the local underwater transmission spectrum is mainly determined by the vertical distribution of iron hydroxide. Therefore, we performed numerical simulations to estimate the vertical distribution of iron hydroxide by considering its formation via chemical reaction, eddy diffusion, and precipitation. As reduced iron, primarily sourced from the ocean floor, reaches the boundary between oxygenic and anoxygenic zones, iron hydroxide is formed via chemical reactions between oxygen and reduced iron. While reduced iron, oxygen, and iron hydroxide are transported solely by vertical diffusion, iron hydroxide also undergoes precipitation. As an initial condition, the calculation box contains oxygen only above the pycnocline. We assumed that the concentrations of oxygen in the oxic region (i.e. above the pycnocline) and reduced iron at the bottom of the box are constant over the calculation time. Reduced iron supplied from the bottom of the box spreads throughout the box due to vertical diffusion over time.

We have also stated in the main text that vertical diffusion is primarily driven by oceanic eddies. These eddies introduce more dynamic particle movements, which may encompass certain physical effects, such as the transportation of iron hydroxide to shorelines. Therefore, we did not assume a negligible vertical diffusion on smaller scales.

L69 in Main Text:

Given the vertical structure of open oceans^{22,23}, iron hydroxide likely spread across cyanobacterial habitats due to high eddy diffusivity above the pycnocline.

L77 in Main Text:

Since the concentration of oxygen remained constant due to the high eddy diffusivity above the pycnocline, iron hydroxide was likely formed at the boundary between the oxygenic and anoxygenic zones and may have continuously spread across the cyanobacterial habitats.

- (2) One question that needs addressing is how do photoferrotrophs fit into this model. The authors believe that prior to cyanobacteria the light window in the photic zone was blue, which meant phycoerythrobilin was not needed. Then, when cyanobacteria facilitated Fe(II) oxidation, ferrihydrite formed and accessory pigments become advantageous in a green light window? But if photoferrotrophs caused BIF since 3.7 Ga, then presumably the light window was already green when cyanobacteria evolved, so is this then an adaptation? It would be helpful if the authors could clarify because it seems the role of photoferrotrophs is not being considered, except briefly mentioned in the final section and in Extended Data Table 1.

RESPONSE: Thank you for your valuable comments, which have enhanced the scientific value of our manuscript. We agree with your suggestion and have thoroughly incorporated the impact of photoferrotrophs on the light window and the evolution of cyanobacteria. We also constructed a big picture of the Archean water environment, including the habitats of cyanobacteria and photoferrotrophs, as shown in Figure 1a. As discussed below, we found that photoferrotrophs could thrive even after the green light window was formed. Based on the above consideration, we have incorporated the following sentences in the main text and Supplementary Information as follows:

L204 in Main Text:

Depending on the evolutionary timing of iron oxidizing bacteria³⁸, photoferrotrophy perhaps induced the green-light window prior to cyanobacterial evolution and thus primitive cyanobacteria were possibly under selective pressure towards green light from early on. Considering that photoferrotrophs can thrive in very low-light environments, it is possible that photoferrotrophs continued to exist in anoxic regions below the pycnocline even after the green light environment had fully formed (refer to Fig. 1a). In fact, at a depth of 50 meters under a 10 μM concentration of iron hydroxide, a very faint light environment ($\sim 0.1 \mu\text{mol}/\text{m}^2/\text{s}/\text{nm}$) might exist, with light levels comparable to those in current green light environments ($\sim 0.01\%$ PAR)³³. Additionally, the weak absorption spectra of bacteriochlorophyll (BChl) a and b, ranging from 550 to 600 nm, are fully overlapped with the green light environment.

L255 in Main Text:

The stepwise development of phycobilisomes seems to mirror the gradual change in the light window due to the oxidation of the aquatic environment by cyanobacteria or the direct oxidation of reduced iron by photoferrotrophs, which may have initiated prior to the birth of cyanobacteria³⁸.

L236 in Supplementary Information:

There are three possible scenarios to form a green-light window during the Archean era, depending on how to oxidize reduced iron supplied from thermal vents and lands: 1. indirect oxidation of reduced iron by oxygenic photosynthesis in cyanobacteria^{25,27}, 2. direct oxidation of reduced iron by photoferrotrophs²⁶⁻²⁸, and 3. photochemical oxidation of reduced iron by UV radiation⁴⁰. In the third scenario, the green-light window could have started forming around the time oceans emerged. In the first and second scenarios, the timing of the green-light environment's formation depends on the appearance of photosynthetic organisms – cyanobacteria and photoferrotrophs.

L247 in Supplementary Information:

Photoferrotrophs are hypothesized to have played an important role in the formation of the initial BIF⁴³, suggesting their emergence prior to the evolution of cyanobacteria. Hence, the green-light window induced by photoferrotrophs (scenario 2) may have been present already in the early Archean, regardless of the significance of scenario 3, which could have induced the green-light window even earlier.

L261 in Supplementary Information:

Overall, while the green light window may have emerged as early as 3.7 billion years ago through the oxidation of reduced iron by photoferrotrophs and/or UV radiation, before the emergence of cyanobacteria, oxygenic photosynthesis further intensified this process from ~3 billion years ago.

Finally, we have considered how photoferrotrophs are fit into the blue-light window and added the following sentences in the discussion section:

L245 in Main Text:

This could explain the alignment of Chl a– and BChl-based photosynthesis with the blue-light window, suggested by the absorption spectrum of Chl a and BChl, matching the spectral range of this window^{46,47}.

- (3) Perhaps it is just my lack of knowledge about phycobilins, but the text gets confusing in terms of discussing the origins of PE (protein) versus PEB (pigment). For instance, L132 where it is mentioned that PEB evolved before PE. But above, L121, it states that: “We further investigated the necessity of the phycobilin pigment PEB that specifically attaches to PE in native hosts. So, if PEB requires the protein to attach, then how could it evolve before the protein? Similarly, L151 states that the PE-PC- APC triad already existed in the common ancestor of crown-group cyanobacteria. This suggests to me then that the pigment already evolved before then, and if so, why?”

RESPONSE: We appreciate your concern. In L121 of the original manuscript, we did not intend to argue that PEB requires PE to be functional. The sentence is only about the necessity of PEB to effectively utilize green light, not about the necessity of a particular pigment that only attach to PE. In fact, PEB has the ability to attach both PE and PC. Based on these considerations, we discussed it in Supplementary Discussion 9 and have added the following sentences in the main Text to prevent the reader's confusion as follows:

L148 in Main Text:

We further investigated the necessity of the phycobilin pigment PEB that specifically attaches to PE in modern cyanobacteria...

L160 in Main Text:

Our observation implies that the utilization of green light does not necessarily require PE, but rather it is sufficient for PEB to simply attach to PC. The evolution of PEB would have promoted the functional or structural specialization of PC, likely resulting in the evolution of PE and ultimately modern phycobilisomes (Supplementary Discussion 9).

- (4) It is unclear to me whether the authors are envisioning a planktonic versus benthic cyanobacterial community. The model species are both benthic, so I presume that means the authors believe the green light window was comprised of microbial mats? But, both species are freshwater, yet the environment in question is coastal?

RESPONSE: Thank you very much for your comment, which provides valuable insight into the scientific merit of this manuscript. In the original manuscript, we did not discuss the selection of model species for the natural selection experiments, as we believed that the habitat of cyanobacteria does not significantly affect the main conclusions of this study. However, we acknowledge your concern that the absence of such an explanation might lead to a confusion for readers. Therefore, we have included a detailed discussion in the main text on how the habitat of cyanobacteria influences the light window and impacts the main conclusion of this study as follows. We have added a new section to Supplementary Information and have extensively discussed it in Supplementary information 6.

L109 in Main Text:

It is important to note that this implication applies to various cyanobacterial habitats, such as open oceans, coastal areas, and freshwater environments. Additionally, because the green light window spreads only a few meters below the surface, both planktonic and benthic cyanobacterial communities would have grown under similar light conditions, assuming that planktonic cyanobacteria were primarily transported by oceanic diffusions, currents and tides. Thus, regardless of the cyanobacterial species and habitats, the coevolutionary relationship between the light window and cyanobacteria can be envisioned (Supplementary Discussion 6).

Specific comments:

- L5: the authors suggest that aerobic biodiversity occurred after the GOE at 2.4 Ga. This is to some extent true, but it fails to recognize that there is a body of published work that proposes aerobic metabolisms already existed since at least 3.1 Ga. Jabłońska, J., Tawfik, D.S. The evolution of oxygen-utilizing enzymes suggests early biosphere oxygenation. *Nat Ecol Evol* 5, 442–448 (2021).

RESPONSE: Thank you for providing the above reference. We have revised the manuscript to incorporate the reference and updated the sentences, weakening the original argument.

L5 in Main Text:

Cyanobacteria induced the Great Oxidation Event (GOE) at ca. 2.4 billion years ago, likely triggering the rise in aerobic biodiversity.

L33 in Main Text:

The GOE likely played an important role in the promotion of aerobic biodiversity³. However, it is worth noting that recent research suggests an emergence of aerobic metabolism and thus an oxygenated before the GOE⁴.

- L36: A reference is needed.

RESPONSE: In accordance with your comment, we have added a reference in L40.

- L63-64. Will this statement hold true if we take some key factors into consideration, such as Fe(II) source as hydrothermal plumes (point sources), upwelling currents, sedimentation of ferrihydrite, as well as water depth? High diffusivity surely makes it easier to evenly distribute dissolved ions, but why would ferrihydrite spread by diffusion versus advection? After all there are currents. Also, do you mean “spread evenly” as surface waters or bottom waters? If the former, the authors need to explain how given their density because we know ferrihydrite sank to the shelf seafloor because of the abundance of banded iron formations at that time.

RESPONSE: Thank you for raising this concern. We agree with your assessment that the above sentence has not been validated because of lateral advection. Therefore, we have removed the following phrases from the main text to address this issue.

L90 in Main Text:

~~*suggesting that iron hydroxide could spread over the entire Earth.*~~

L69 in Main Text:

Given the vertical structure of open oceans^{22,23}, iron hydroxide likely spread evenly across cyanobacterial habitats due to high eddy diffusivity above the pycnocline.

- L74: similar to the comment above, why would “iron hydroxide spread over the entire Earth”. This implies that the ferric mineral stayed buoyant?

RESPONSE: Instead of the above deleted phrases, we described the possibility of small iron hydroxide particles remaining buoyant immediately after its formation as follows:

L84 in Main Text:

Based on our experiments on the iron hydroxide formation (Extended Data Fig. 2), the particle size of iron hydroxide was likely small (< 100 nm) immediately after formation, allowing it to remain buoyant in short periods. As a result, iron hydroxide would have consistently influenced the underwater light environment in the cyanobacterial habitat as long as reduced iron was oxidized through photoferrotrophic and cyanobacterial activity...

- L76-79. The molar absorption coefficient can be indirectly affected by particle size and thus change the path length. In this case, this is especially true due to the fact that ferrihydrite particles are (1) constantly changing in size – see ref below, (2) aggregation by ionic bridging effect or disaggregation by physical forces, and (3) morphology and crystallinity change drastically among different synthesis method. All these characteristics of ferrihydrite affect the electronic structure and transition probabilities of the molecules and thus the light absorption.
Li, Y., Sutherland, B.R., Gingras, M.K., Owttrim, G.W., and Konhauser, K.O., 2021. A novel approach to investigate the deposition of (bio)chemical sediments: The sedimentation velocity of cyanobacteria-ferrihydrite aggregates. *Journal of Sedimentary Research*, 91:390-398.
- Method L421: The two ferrihydrite synthesis methods utilized in this study do not accurately represent the actual oxidation process of Fe(II) by cyanobacteria and photoferrotrophs, which could drastically change the particle size, morphology, and crystallinity of minerals particles. It should be considered if this discrepancy could cause a change in light window.

RESPONSE: We appreciate your insightful comments. We agree with your assessments and understand them. Based on your concern, we have performed additional measurements regarding the relationship between particle size and the molar absorption coefficient of iron hydroxide. As shown in Extended Data Figure 2, the molar absorption coefficient is nearly independent of particle size in the visible wavelength range. However, we were not unable to evaluate the dependence of aggregation and morphology on the coefficients. Therefore, we have mentioned the impact of these factors on the light window in the main text and Supplementary Information as follows. We have also discussed the uncertainty of the light window calculation in Supplementary Discussion 5.

L94 in Main Text

While different types of iron hydroxide might have existed during the Archean era²⁸, only the earliest phase of small-sized iron particles likely had the most prolonged impact on the underwater light environment. (see also Supplementary Discussion 5).

L174 in Supplementary Information

Second, only iron hydroxide was considered in this study because small particles, just after formation, remain buoyant and affect the underwater transmission spectrum for an extended period based on the Stokes' law. As shown in Extended Data Figure 2, the molar absorption coefficient is almost independent of particle size in the visible wavelength range. However, the molar absorption coefficient also depends on aggregation with cyanobacteria³⁰, particle morphology, and crystallinity³¹. Additionally, various types of ferrihydrite particles likely contributed to the formation of BIFs³¹, leading to uncertainties in the absorption coefficient even in the visible range. Therefore, the impact of different conditions and types of ferrihydrite particles on the underwater transmission spectrum should be investigated in future studies.

- L100-102. The pycnocline refers to a zone in oceans that have a significant change in density, due to temperature, salinity and turbidity. It is not the same as the photic zone. But this text makes it seem like they are equivalent.

RESPONSE: Thank you for pointing this out. We entirely removed the reference to photic zone in the main text. Instead, we have used the following phrase, “*at the boundary between the oxygenic and anoxygenic zones*” in the main text and have clearly described the assumption that the pycnocline corresponds to the boundary between these two zones:

L75 in Main Text:

Based on the previous studies^{17,26}, we assumed that the oxygenic and anoxygenic zones are divided by the pycnocline. Since the concentration of oxygen remained constant due to the high eddy diffusivity above the pycnocline, iron hydroxide was likely formed at the boundary between the oxygenic and anoxygenic zones and may have continuously spread across the cyanobacterial habitats.

L125 in Main Text:

*In conclusion, an underwater green-light environment during the Archean era was highly plausible, as reduced iron supplied from hydrothermal vents was continuously oxidized **at the boundary between oxidized and reduced environments**, leading to the formation of BIFs.*

- L111-112: Please explain why you chose these two cyanobacterial species. I get *Gloeobacter* as it is the deepest cyanobacteria lineage, but I think it is a terrestrial endolith. *Synechococcus elongatus* is also freshwater, unicellular, mat-forming and evolved in the Mesoproterozoic. In both cases you are using benthic species versus planktonic. I bring this up because it is not clear in the manuscript what mode of growth you are assuming for the Archean, i.e., mats or plankton. It would also be useful to mention somewhere that these are extant species, so we have no way of knowing for sure whether they represent the species that might have dominated Archean seawater.

RESPONSE: We are thankful for your suggestion. We have added a justification for the selection of model species for natural selection experiments and its impact on the main conclusion of our study, including the following sentences in Main Text and Supplementary Information as follows:

L654 in Main Text:

These are extant species in the point of divergence time and habitat, so it remains uncertainty whether they accurately represent species that might have dominated in the Archean era. However, because the light window analysis can be applied to various species and habitats, the selection of model species does not affect the main conclusions regarding the coevolutionary relationship between the light environment and cyanobacteria (Supplementary Discussion 6).

L225 in Supplementary Discussion 6:

*We selected *Gloeobacter violaceus* PCC7421 and *Synechococcus elongatus* PCC 7942 as model species. While these species are extant and are defined by their divergence time and habitat, it remains uncertain if they represent species that dominated during the Archean era. However, as*

discussed above, the choice of cyanobacteria as model species does not affect the conclusions of this study.

- L169-174: Ferruginous conditions were commonplace in early Archean so why not start out with green light, especially if photoferrotrophs were already in existence?
- L215-217. Photoferrotrophs and cyanobacteria did not evolve around the same time. Photoferrotrophs alone are shown to be capable of forming massive scale BIFs, which inherently infers that there was a significant amount of ferrihydrite particles in seawater before cyanobacteria evolved. “This pivotal transition” likely happened sooner than authors state here.
Konhauser, K.O., Hamade, T., Morris, R.C., Ferris, F.G., Southam, G., Raiswell, R., and Canfield, D., 2002. Could bacteria have formed the Precambrian banded iron formations? *Geology*, 30:1079- 1082.

RESPONSE: We value your suggestion. We have discussed a scenario in which the green light environment may have formed due to the direct oxidation of reduced iron by photoferrotrophs before the emergence of cyanobacteria in Main Text and Supplementary Information, citing the above reference.

L204 in Main Text:

Depending on the evolutionary timing of iron oxidizing bacteria³⁸, photoferrotrophy perhaps induced the green-light window prior to cyanobacterial evolution and thus primitive cyanobacteria were possibly under selective pressure towards green light from early on. Considering that photoferrotrophs can thrive in very low-light environments, it is possible that photoferrotrophs continued to exist in anoxic regions below the pycnocline even after the green light environment had fully formed (refer to Fig. 1a). In fact, at a depth of 50 meters under a 10 μM concentration of iron hydroxide, a very faint light environment ($\sim 0.1 \mu\text{mol}/\text{m}^2/\text{s}/\text{nm}$) might exist, with light levels comparable to those in current green light environments ($\sim 0.01\%$ PAR)³³. Additionally, the weak absorption spectra of bacteriochlorophyll (BChl) a and b, ranging from 550 to 600 nm, are fully overlapped with the green light environment.

L255 in Main Text:

The stepwise development of phycobilisomes seems to mirror the gradual change in the light window due to the oxidation of the aquatic environment by cyanobacteria or the direct oxidation of reduced iron by photoferrotrophs, which may have initiated prior to the birth of cyanobacteria³⁸.

L265 in Supplementary Information:

Photoferrotrophs are hypothesized to have played an important role in the formation of the initial BIF⁴³, suggesting their emergence prior to the evolution of cyanobacteria. Hence, the green-light window induced by photoferrotrophs (scenario 2) may have been present already in the early Archean, regardless of the significance of scenario 3, which could have induced the green-light window even earlier.

L272 in Supplementary Information:

Overall, while the green light window may have emerged as early as 3.7 billion years ago through the oxidation of reduced iron by photoferrotrophs and/or UV radiation, before the emergence of cyanobacteria, oxygenic photosynthesis further intensified this process from ~3 billion years ago.

- L210-212: Although this makes sense, I think it is a bit too simplistic because the explanation here does not include the possibility of high concentrations of ferrihydrite particles formed during anoxygenic photosynthesis as early as in the Paleoproterozoic, and suspended clays from continental input (as evidenced by shale deposits). If and how these factors affect the light window is not discussed here but should be. Even in a world where Fe(II)-oxidizing microbes did not yet evolve, Fe(II) could have been oxidized photochemically via UV light and/or Fe(II)-silicates could have formed at high concentrations. How does that affect the proposed “blue light window”.
- Konhauser, K.O., Amskold, L., Lalonde, S.V., Posth, N.R., Kappler, A., and Anbar, A., 2007. Decoupling photochemical Fe(II) oxidation from shallow-water BIF deposition. *Earth and Planetary Science Letters*, 258:87-100.

RESPONSE: Thank you also for your insightful comment. In order to address your concern, we have discussed the possibility of the early emergence of a green light environment due to photochemical oxidation of reduced iron by UV radiation in the discussion section, citing the above reference.

L258 in Main Text:

A green light environment may have additionally been created through the photochemical oxidation of reduced iron (Fe(II)) by UV radiation^{48,49}. Resulting Fe(III)-rich minerals might have also protected cyanobacterial habitat from harmful UV radiation (Extended Data Fig. 2).

We have also summarized how the green light environments have formed during the Archean era in Supplementary Discussion 7 as follows:

L233 in Supplementary Information:

There are three possible scenarios to form a green-light window during the Archean era, depending on how to oxidize reduced iron supplied from thermal vents and lands: 1. indirect oxidation of reduced iron by oxygenic photosynthesis in cyanobacteria^{25,27}, 2. direct oxidation of reduced iron by photoferrotrophs²⁶⁻²⁸, and 3. photochemical oxidation of reduced iron by UV radiation⁴⁰. In the third scenario, the green-light window could have started forming around the time oceans emerged.

- Final section: I think this is a great summary but it is only here that photoferrotrophs are mentioned in passing.

RESPONSE: Thank you for raising your concern. We have considered how photoferrotrophs fit into this scenario and have discussed it in the discussion section as follows.

L240 in Main Text:

Because of the nearly anoxic atmosphere before the GOE⁴³, Earth's surface was potentially exposed to harmful UV-C light longer than 200 nm⁴⁴. Primitive cyanobacteria and photoferrotrophs likely thrived in deeper regions of the photic zone (~ 20 – 30 meters)²⁸ due to the lack of protective shields against UV-C radiation⁴⁵. The light window, predominantly influenced by water, favored a spectrum leaning towards blue, hence termed the "blue-light window" (left panel of Fig. 4). This could explain the alignment of Chl a- and BChl-based photosynthesis with the blue-light window, suggested by the absorption spectrum of Chl a and BChl, matching the spectral range of this window^{46,47}.

L248 in Main Text:

The stepwise development of phycobilisomes seems to mirror the gradual change in the light window due to the oxidation of the aquatic environment by cyanobacteria or the direct oxidation of reduced iron by photoferrotrophs, which may have initiated prior to the birth of cyanobacteria³⁸.

- Figure 1a: Dissolved silica could easily keep a significant amount of Fe(II) in solution in the presence of oxygen. Organic ligands derived from microbes have the same effect.

RESPONSE: We have discussed the impact of dissolved silica on the concentration of iron hydroxide in the method of Main Text and Supplementary Discussion 5 as follows:

L514 in Main Text:

It is also important to note that silica might have been largely dissolved in the Archean ocean²¹, leading to a much higher concentration of reduced iron even in the oxidized zone. This is because the dissolved silica is transformed into silica-Fe(II) minerals through reaction with reduced iron⁶². As a result, the concentration of iron hydroxide may have been affected by silica. However, because it is difficult to quantitatively evaluate the reaction rate between dissolved silica and reduced iron, we did not consider the impact of the dissolved silica on the underwater transmission spectrum, which should be treated by a future study (see also Supplementary Discussion 5).

L187 in Supplementary Information

On the other hand, dissolved silica may have been abundant in the Archean Ocean²⁸, potentially leading to the formation of silica-Fe(II) minerals through chemical reaction between silica and reduced iron. The presence of dissolved silica could enhance the concentration of reduced iron even under oxidized conditions³³, thereby reducing the concentration of iron hydroxide.

- Also, a concentration of 100 μM Fe is only a conservative estimate of the Archean seawater Fe concentration. This concentration can be as high as 500 μM , which would cause high turbidity. Ideally, this range of Fe concentration should be modelled instead of a fixed concentration of 100 μM .

RESPONSE: Thank you for bringing this to our attention. We have performed numerical simulations for ten times lower and higher concentrations of iron hydroxide as extreme cases and have added Extended Data Figure 1g – h and modified Extended Data Figure 3c and Extended

Data Figure 4k. We have also discussed the assumptions and results in Supplementary Discussion 2 as follows:

L68 in Supplementary Discussion:

Additionally, considering that there is uncertainty in the concentration of reduced iron, Models 7 and 8 apply ten times lower and higher than the standard value of 80 μM as an extreme case, which are observed in the Archean ocean analogues¹.

L73 in Supplementary Discussion:

The concentration of iron hydroxide in the mixed layer, which serves as the habitat for cyanobacteria, is mainly determined by the influx of reduced iron to the pycnocline. The value varies between 1 and 10 μM under realistic conditions and increases up to 100 μM as an extreme case.

L134 in Supplementary Discussion:

The right side of Extended Data Fig. 3 depicts the photon flux density at this shallower habitat depth. Predominantly, the light window ranges between 500 and 600 nm wavelengths, except under modern ocean conditions (without the presence of iron hydroxide) and in high turbidity conditions, where the maximum iron hydroxide concentration reaches 100 μM . While the modern ocean provides a broad wavelength window ranging from blue to green light, only faint red light is available near the water surface under such high turbidity.

Dear Reviewer,

Thank you very much for evaluating our study and for providing such insightful and constructive comments. Your feedback has significantly improved the scientific quality of our manuscript.

We have also received a number of useful comments from the other reviewers. We have carefully considered all the reviewer's suggestions and revised the manuscript. All revisions in both the main text and supplementary information are highlighted in yellow. Before detailing our responses to each comment, we would like to provide an overview of the major revisions made to the main text and supplementary information. A summary of the major revisions is as follows:

1. We have expanded our discussion in both the main text and supplementary information on how the green light environment—induced by the direct oxidation of reduced iron by photoferrotrophs— could influence the coevolutionary relationship between cyanobacteria and their light environment.
2. The assumptions for our numerical calculations on the distribution of iron hydroxide have been extensively clarified in the methods section. Additionally, we have added a panel (Figure 1a) illustrating the water environment during the Archean era.
3. We have incorporated a detailed discussion on the potential adaptation of cyanobacteria to white light environments after the two oxidation events.
4. A new section has been added to the Supplementary Information to address uncertainties in the light window analysis during the Archean era (Supplementary Discussion 5).
5. We have also discussed the influence of cyanobacterial habitat on our main conclusions in another newly added section of the Supplementary Information (Supplementary Discussion 6).
6. Figure 3c, which shows the fraction of species bearing PE as a function of time, has been revised to account for uncertainties in previous molecular clock analyses. The corresponding method section has also been updated.
7. We have entirely updated our phylogenetic analyses, replacing Figure 3, Extended Data Figure 9, and their source files with the new results.
8. Lastly, we have made minor grammatical adjustment throughout the manuscript to improve readability.

We have addressed each comment from you in detail below. Our responses, as well as the newly inserted or modified sentences of the main text and supplementary information, are highlighted in blue. We hope these revisions meet your expectation.

Once again, we sincerely appreciate your time and efforts in reviewing this study.

Sincerely yours,

---- Response to each comment ----

1. Terrestrial cyanobacteria can be added to the discussion. Several cyanobacteria in the phylogeny analyses (Source data for Figure 1a, 3b, and Extended Data Figure 9) are terrestrial cyanobacteria, such as *Gloeobacter*, *Anthocerotibacter*, *Leptolyngbya*, *Chroococcidiopsis*, and *Nostoc*. The authors briefly discuss this in Supplementary Discussion 6. However, it would be nice to include the idea in Figure 4 that terrestrial cyanobacteria thrive in the white light spectra on land.

RESPONSE: Thank you for your insightful comment. Although we may not correctly understand your intention, this discussion offers valuable insights to strengthen our manuscript by highlighting the adaptation of cyanobacteria to the white light environment on land following the GOE. We have extensively discussed this point as follows:

L273 in Main Text:

Terrestrial cyanobacteria then successfully spread over the entire Earth by adapting to various land conditions⁹. Compared to marine cyanobacteria that thrive under various light environments, including dominant green light environment, terrestrial cyanobacteria tend to possess only APC and PC⁹; this is possibly linked to the loss of PE as an adaptation to white light environments on land. In other words, whereas PE becomes less important in brighter white light conditions on land, it would play a more crucial role in dimmer underwater conditions. From the perspective of energy transfer from the light-harvesting antenna to the photosystem, this adaptation suggests that cyanobacteria may have favored to employ pigments absorbing light at wavelengths close to those absorbed by the special pair of photosystems.

2. The authors can discuss the widespread of *Prochlorococcus* in the modern ocean. This strain has divinyl Chl a and Chl b and no phycobilisome, which differs from other crown cyanobacteria in pigment composition. The different pigment composition might be an adaptation to the modern ocean's white light window.

RESPONSE: We appreciate your suggestion to include *Prochlorococcus* as a potential example of cyanobacterial adaptation to the white light window. After careful consideration, we have decided not to include *Prochlorococcus* in this study. This decision is based on the understanding that the loss of phycobilisomes might not be driven solely by the underwater light environment but also by nitrogen availability. On the other hand, based on the recent study (Ulloa et al. 2021), we have also noticed a potential link between the origin of *Prochlorococcus* and the oxidation of the aquatic environment during the second major oxidation event, the Neoproterozoic Oxygenation Event (NOE), which led to the formation of the blue light window due to the full oxidation of the oceans. Your comment has encouraged us to further investigate the potential coevolutionary relationship between cyanobacteria and the light environment during the NOE in future studies.

Reference: Ulloa, O. Henríquez-Castilloa, C. Ramírez-Flandes, S. et al. The cyanobacterium *Prochlorococcus* has divergent light-harvesting antennae and may have evolved in a low-oxygen ocean. *Proc. Natl. Acad. Sci. USA*. **118**, e2025638118 (2021).

3. Line 501 and 511: “a” in chlorophyll a should be italicized.

RESPONSE: Thank you for checking our manuscript in detail. We have made the necessary correction.

- 3. Line 517: Cyanobacterial strains and culture conditions. Please describe the source of these two cyanobacterial strains (purchased from a culture collection center or gifted by other labs?)

RESPONSE: Thank you also for your suggestion. We have described the source of cyanobacterial strains and culture conditions in the method section as follows:

L660 in Main Text:

The cultures of G. violaceus PCC7421 were purchased from the Pasteur Culture Collection (PCC), while the cultures of S. elongatus PCC 7942 was prepared from our own collection in Nagoya University.

- 4. Line 518: Based on the recent classification, Gloeobacteraceae and Anthocerotibacteraceae were classified into the order Gloeobacterales. In the earliest-branching clade (Gloeobacterales), Gloeobacter only represents the Gloeobacteraceae but not Anthocerotibacteraceae because these two clades diverged 1.4 billion years ago and have distinct features (Anthocerotibacter does not have PE and have paddle-shaped phycobilisomes).

RESPONSE: Thank you as well for providing this valuable information. We have added the following statement to the methods section:

L635 in Main Text:

It is important to note that Gloeobacter only represents Gloeobacteraceae, while Anthocerotibacteraceae is also classified into the order Gloeobacterales⁷⁵. These two clades diverged 1.4 billion years ago and have distinct features: Anthocerotibacter lacks PE and have paddle-shaped phycobilisomes⁷⁶.

- These two references can be added in this paragraph.
Strunecký O, Ivanova AP, Mareš J (2023) An updated classification of cyanobacterial orders and families based on phylogenomic and polyphasic analysis. J Phycol 59 (1):12-51.
Rahmatpour N, Hauser DA, Nelson JM, Chen PY, Villarreal A JC, Ho M-Y, Li F-W (2021) A novel thylakoid-less isolate fills a billion-year gap in the evolution of Cyanobacteria. Curr Biol 31 (13):2857–2867.e2854.

RESPONSE: According to your suggestion, we have added the above references in L650 and 651, respectively.

- *Gloeobacter violaceus* PCC 7421 was isolated from calcareous rock in Switzerland, not from the ocean. As the authors' model of cyanobacteria evolution is in the ocean, *Gloeobacter violaceus* PCC 7421 might not be an ideal model organism to represent early-diverging cyanobacteria in the ocean.

RESPONSE: We appreciate your concern and fully understand it. Since we initially believed that the habitat of cyanobacteria did not directly affect the main conclusions of this study, we did not specify the assumptions regarding the habitat in our model. However, recognizing that the absence of such a description might lead to further confusion for readers, we have thoroughly discussed the relationship between the habitat of cyanobacteria and the light window in Supplementary Discussion 6.

- 5. Line 528: Please justify using two different wavelengths (750 nm and 730 nm) to measure the optical densities of *G. violaceus* and *S. elongatus*.

RESPONSE: The wavelength to measure OD of cyanobacteria is often 730 nm or 750 nm, although it varies from laboratory. Both wavelengths measure the light scattering of cells without the effect of absorption by Chl *a*, so the values of OD do not change significantly. OD of *Synechococcus elongatus* PCC 7942 was measured using 730 nm, because we have maintained this strain for more than 30 years and have used this wavelength in the laboratory. On the other hand, we purchased *Gloeobacter* for this research. OD of this strain was measured using 750 nm, because this wavelength was used for OD measurement of *Gloeobacter* in the laboratory of coauthor Hideaki Miyashita. We have added a sentence below.

L673 in Main Text:

Both wavelengths measure light scattering by cells without being affected by Chl a absorption, so the OD values do not change largely.

- 6. Source data for Figure 3b and Extended Data Figure 9: *Anthocerotibacter panamensis* does not have CpcC. The CpcC sequence of *Anthocerotibacter panamensis* used in the phylogeny has an updated annotation as CpcJ based on a recent publication last year. Please refer to the following reference: Jiang H-W, Wu H-Y, Wang C-H, Yang C-H, Ko J-T, Ho H-C, Tsai M-D, Bryant DA, Li F-W, Ho M-C, Ho M-Y (2023) A structure of the relict phycobilisome from a thylakoid-free cyanobacterium. *Nat Commun* 14 (1):8009.

RESPONSE: Thank you for your suggestion. We have modified the annotation for *Anthocerotibacter* in the source data of Figure 3b and Extended Data Figure 9. We have also included a brief statement in the method section as follows:

L732 in Main Text:

It is noted that the CpcC homolog in Anthocerotibacter is now annotated as CpcJ⁸⁰.

Dear Reviewer,

Thank you very much for evaluating our study and for providing such insightful and constructive comments. Your feedback has significantly improved the scientific quality of our manuscript.

We have also received a number of useful comments from the other reviewers. We have carefully considered all the reviewer's suggestions and revised the manuscript. All revisions in both the main text and supplementary information are highlighted in yellow. Before detailing our responses to each comment, we would like to provide an overview of the major revisions made to the main text and supplementary information. A summary of the major revisions is as follows:

1. We have expanded our discussion in both the main text and supplementary information on how the green light environment—induced by the direct oxidation of reduced iron by photoferrotrophs— could influence the coevolutionary relationship between cyanobacteria and their light environment.
2. The assumptions for our numerical calculations on the distribution of iron hydroxide have been extensively clarified in the methods section. Additionally, we have added a panel (Figure 1a) illustrating the water environment during the Archean era.
3. We have incorporated a detailed discussion on the potential adaptation of cyanobacteria to white light environments after the two oxidation events.
4. A new section has been added to the Supplementary Information to address uncertainties in the light window analysis during the Archean era (Supplementary Discussion 5).
5. We have also discussed the influence of cyanobacterial habitat on our main conclusions in another newly added section of the Supplementary Information (Supplementary Discussion 6).
6. Figure 3c, which shows the fraction of species bearing PE as a function of time, has been revised to account for uncertainties in previous molecular clock analyses. The corresponding method section has also been updated.
7. We have entirely updated our phylogenetic analyses, replacing Figure 3, Extended Data Figure 9, and their source files with the new results.
8. Lastly, we have made minor grammatical adjustment throughout the manuscript to improve readability.

We have addressed each comment from you in detail below. Our responses, as well as the newly inserted or modified sentences of the main text and supplementary information, are highlighted in blue. We hope these revisions meet your expectation.

Once again, we sincerely appreciate your time and efforts in reviewing this study.

Sincerely yours,

---- Response to each comment ----

Major points:

- Could the simulation and calculation of light wavelength and light windows in different depths reflect the real situation of Archean aquatic world? Different pH, ion composition, atmospheric conditions and biotic environment dynamically controlled the Archean aquatic light condition. Please provide some explanations.

RESPONSE: Thank you very much for your valuable comment. We agree that there were not comments on the uncertainty of the light window analysis in the original material. Therefore, we have added a new section to Supplementary Information and have explained the uncertainty of the calculation of light windows due to different pH, ion composition, and atmospheric conditions in Supplementary Discussion 5. We have also added an assumption of biotic environment in the Archean era in the method section on the calculation of light window as follows:

L551 in Main Text:

In modern water environments, the absorptions of water, dissolved ion, phytoplankton, non-algal particles, and colored dissolved organic matters primarily govern the underwater transmission spectrum⁶⁴. However, given that biomass might be much lower in the Archean Ocean than in modern water environments due to very limited availability of phosphorus⁶⁵, the absorptions of phytoplankton and colored dissolved organic matters were ignored. It is also important to note that, although reduced iron was abundant in the Archean water environments, its absorption coefficient is very low except for the UV wavelength range⁶⁶. As a result, the effects of only water and iron hydroxide as non-algal particles on the underwater transmission spectrum were considered. On the other hand, there may also exist some other factors affecting the light window. The uncertainties related to the light window analysis are further discussed in Supplementary discussion 5.

- 2. Genetic engineering on *Synechocystis* sp. PCC6803 resulted in the green-light adaptation of this no-PE cyanobacteria. But the growth of transgenic cyanobacteria changed the color of water and culture medium. Will the changes influence the results of genetic engineering simulation of PE-contained *Synechocystis* sp. PCC6803?

RESPONSE: Thank you also for your insight regarding our experiments. We believe that cyanobacterial abundance had minimal impact on the color of the water, as the cell cultures were consistently stirred by bubbling air throughout the experiments. This condition is consistent with the following assumption. Additionally, considering the limited availability of phosphorus during the Archean era, cyanobacterial abundance would not have been high enough to cause significant self-shielding. Thus, we assumed that cyanobacteria were not present in sufficient quantities to affect the colour of the water. To clarify this point, we have added the following sentence to the methods section as follows:

L668 in Main Text:

It is important to note that the cell cultures stirred by bubbling with air during the experiments had no impact on the transmission spectrum. This observation aligns with the assumption that cyanobacterial abundance might have been low in the Archean era due to the limited availability of phosphorus⁶⁵.

- 3. I am very curious that green light occupied a large area of visible spectrum and have higher energy, but why did cyanobacteria and almost all the photosynthetic organisms choose the red light rather than the green light as one of the main energy sources?

RESPONSE: First of all, a previous study suggests that approximately 80% of green light is utilized for photosynthesis by higher plants from a structural perspective of leaves (Terashima et al., 2009). This indicates that green light remains an important energy source for certain photosynthetic organisms. In contrast, photosynthetic bacteria, including cyanobacteria, do not efficiently absorb green light in the absence of PE. Therefore, we have focused on the question of why most cyanobacteria do not use green light effectively for photosynthesis. Providing a clear explanation for this fundamental question is challenging based solely on our current analysis and understanding. However, we hypothesize that Chl *a* is incorporated into the special pair of the photosystem because it allows access to the entire visible spectrum, which is essential for water oxidation in oxygenic photosynthesis. From the perspective of energy transfer within the phycobilisome to the photosystem, photosynthetic organisms tend to favour pigments that absorb light at wavelengths similar to those of Chl *a* within their light-harvesting antennas. In other words, it is more difficult for cyanobacteria to utilize green light efficiently for photosynthesis. Based on these considerations, we have added the following sentences to the discussion section:

L277 in Main Text:

In other words, whereas PE becomes less important in brighter white light conditions on land, it would play a more crucial role in dimmer underwater conditions. From the perspective of energy transfer from the light-harvesting antenna to the photosystem, this adaptation suggests that cyanobacteria may have favored to employ pigments absorbing light at wavelengths close to those absorbed by the special pair of photosystems.

Reference: Terashima, I. Fujita, T. Inoue, T. et al. Green light drives leaf photosynthesis more efficiently than red light in strong white light: Revisiting the enigmatic question of why leaves are green. *Plant Cell Physiol.* **50**, 684 – 697 (2009).

- 4. For the phylogenetic analyses, it is more convincing to add some analyses or discussion for the divergence, duplication and gain-and-loss events of gene families, rather than simple deduction. For example, in line 156: Given the importance of the PE gene, the discussion of how to evaluate the loss of the PE gene in many late-diverging species is missing here.

RESPONSE: We appreciate your insightful comment. We indeed discussed why PE is not universally conserved in modern cyanobacteria in Supplementary Discussion 8. We have added some statements about the driving force of PE loss in the discussion section as follows:

L269 in Main Text:

Following the two oxidation events of Earth, the GOE and the Neoproterozoic Oxygenation Event (NOE), the fully oxidized aquatic environment expanded its light window to white light (right panel of Fig. 4). This transformation resulted from the depletion of oxidized iron species within the aquatic environment and the formation of the ozone layer, thus opening diverse color niches for phototrophic organisms¹⁰. Terrestrial cyanobacteria then successfully spread over the entire Earth by adapting to various land conditions⁹. Compared to marine cyanobacteria that thrive under various light environments, including dominant green light environment, terrestrial cyanobacteria tend to possess only APC and PC⁹; this is possibly linked to the loss of PE as an adaptation to white light environments on land.

In short, the APC-PC-PE system can collectively adapt to a wide range of wavelengths beyond solely green light regions. The adaptation of post-GOE cyanobacteria towards white-light environments (either marine or non-marine), after having been escaped from the green-light burden, let those late comers to delete PE. Yet, in marine settings where green light remains prevalent, many modern cyanobacteria retain PE (Chen et al. 2023).

Reference: Chen et al., Comparative genomics reveals insights into cyanobacterial evolution and habitat adaptation. *ISME Journal* 15:211–227 (2023)

Minor points:

- Line 592: There doesn't seem to be enough details provided here to allow for reproducibility (how was the trimming, masking, etc, done?)

RESPONSE: The amino acid sites that are retained between less than 5% of the analyzed species were removed. We modified the sentence as follows:

L724 in Main Text:

The amino acid sites that are conserved in less than 5% of the analyzed species were removed, except the C- and N-termini, where amino acid sites with < ~50% conservation were removed.

- 2. Line 596: Except the ultrafast bootstrap (UFBOOT), it would be useful to add the branch supports using SH approximate likelihood ratio (SH-aLRT test) approach.

RESPONSE: We appreciate your valuable comment. We have re-run the entire phylogenetic analyses and have added the SH-aLRT values in all figures. The tree topology that is necessary to support our hypothesis is now more robust. We have added the following sentence to the method section.

L741 in Main Text:

Branch support was obtained by Ultrafast bootstrap and Shimodaira-Hasegawa approximate likelihood ratio (SH-aLRT) test in IQ-TREE.

- 3. Line 602-603: What are the criteria for selecting representative sequences?

RESPONSE: The sequences that had the shortest branch length in individual clades were selected. In order to address your concern, we have included the following sentence in the method section:

L736 in Main Text:

By contrast, if those homologs were distributed in the same clades that contain only one species, only the homolog that had the shortest branch length was retained as the representative for the clade.

- 4. Line 628-629: why a previously published molecular clock analysis is being used here, this time frame is robust? Perhaps using your dataset to infer divergence time is better.

RESPONSE: Thank you for your suggestion. While estimating the accurate divergence time of cyanobacteria was indeed one of available options, the dating of cyanobacteria is itself a great controversy in the relevant community. Hence, the accurate dating of cyanobacteria is beyond our scope. There are already several previous studies that comprehensively analyzed various cyanobacterial species. Several general implications are shared among these studies, including the divergence of crown-group cyanobacteria before the GOE, while the divergence time of some specific taxonomic groups are not. Therefore, we chose two different previous studies on molecular clock analysis, conducted by two independent labs using different calibration strategies (Boden et al. 2021; Fournier et al. 2021). We calculated the ratio of PE-bearing cyanobacteria for those who branched off before, during and after the GOE, based on these two studies (Fig. 3c). Their analyses are consistent for the divergence time of early-diverging lineages, which branched before the GOE. Hence, our calculations align well with our hypothesis that the importance of PE for cyanobacteria may have increased due to the green light environment during the Archean era. We have modified Figure 3c and fully revised the method section as follows:

L764 in Main Text:

*The fraction of cyanobacteria having PE that branched before and during the Great Oxidation Event (GOE) was estimated by comparing the distribution of the PE gene (*cpeA*)^{8,9} in the phylum, using a previously published species tree and Bayesian relaxed molecular clock analysis of cyanobacteria^{14,15}. First, we grouped the species listed in two distinct trees^{8,9} in terms of the similarity of 16S rRNA⁸⁵. The 16S rRNA gene sequences of cyanobacterial strains were obtained using the BLAST-N search program in the National Center for Biotechnology Information database (<http://www.ncbi.nlm.nih.gov>). *If the 16S rRNA gene sequence information was unavailable, the respective species were excluded from our samples.* The 16S rRNA gene identity for two adjacent strains in Fig. 3 in Ref. 8 and Fig. 5 in Ref. 9 was calculated with the BLAST-N search program or with ClustalW using the web platform (https://npsa-prabi.ibcp.fr/cgi-bin/npsa_automat.pl?page=/NPSA/npsa_clustalwan.html). Species having >95.0% similarity to each other were grouped in the same species cluster. *By doing so, our calculation remains unaffected by the sampling number of species.* If a cluster contained both species that have and lack *cpeA*, encoding the PE a subunit, the ratio of the species having *cpeA* was calculated as the fraction of the number of the species with *cpeA* to the total number in the cluster. *Second, we**

selected the species that branched before the GOE based on the two independent molecular clock analyses^{14,15}. The taxonomic groups of *Acaryochloris marina* MBIC11017, *Synechococcus* sp. PCC 6312, and *Thermotichus lividus* PCC 6715 were found to have branched just after the GOE. Therefore, we prepared two sets of the early-branching species: one included these groups (brown bar in Fig. 3c), while the other excluded them (green bar). Finally, we calculated the fraction of cyanobacteria having *cpeA* relative to the total number of species for both early-branching species and all extant species.

References:

Boden, J. S., Konhauser, K. O., Robbins, L. J. & Sánchez-Baracaldo, P. Timing the evolution of antioxidant enzymes in cyanobacteria. *Nat. Commun.*, **12**, 4742 (2021).

Fournier, G. P. et al. The Archean origin of oxygenic photosynthesis and extant cyanobacterial lineages. *Proc. R. Soc. B* **288**, 20210675 (2021).

- 5. Figure 3 represents the phylogenetic analysis of phycobiliproteins, bilin biosynthetic enzymes and bilin lyase. What information does the Rod linker phylogenetic tree in Figure 3b intend to convey? It does not match the description in lines 606-624. The information given in the figure legend of Figure 3 is limited, making it rather difficult to understand. Please add more specific information.

RESPONSE: We acknowledge that rod linker was not clearly included in the summary statement of the figure caption and hence modified it in the revised manuscript. Rod linker trees represent the sister relationship of PC- and PE-associated proteins, similar to phycobiliproteins, bilin biosynthetic enzymes and bilin lyases. The revised figure caption is as follows:

Caption of Figure 3b:

Maximum likelihood phylogeny of other phycobilisome-associated proteins: phycobilin synthase, rod linker (connecting different phycobiliproteins) and lyase (conjugating phycobilins with phycobiliproteins). The subclade collapse in each tree is based on the distribution of corresponding paralogs in host cyanobacteria. Light gray indicates the outgroup or proteins that are not part of phycobilisomes. The green and orange colors represent the absorption wavelengths of PE- and PC-associated proteins, respectively.

- 6. Line 603-605: I do not understand why are the sequences outside the clades that contain known PE and PC proteins excluded, but the outgroup sequences, non-cyanobacterial PcyA/PebAB homologs and unknown CpcS Homologs are exceptions.

RESPONSE: We acknowledge that the original statement was confusing and thus modified it in the revised manuscript. Our intention was that our phylogenetic analyses do not aim to elucidate the evolutionary history of the entire protein family, but only focus on the evolution of phycobilisome-related proteins. For instance, phycobiliproteins (APC, PC and PE) have distantly homologous proteins, which probably do not involve in photosynthesis, as suggested by Rockwell et al., 2023. These protein sequences were initially included in the dataset as their phylogenetic positions were not clear, but were eventually mostly excluded. However, some of those sequences were kept as the outgroup.

Main Text in L750:

Distantly related proteins that did not constitute phycobilisomes were used as the outgroup. In the case of phycobilin synthase (PcyA and PebAB) and lyase (CpcS and CpeS). In the case of CpcS, it was not clear which homologs were in fact involved in phycobilisome formation, all clades were colored in orange (Fig. 3b).

Reference:

Rockwell, N. C. et al. Elucidating the origins of phycocyanobilin biosynthesis and phycobiliproteins. *Proc. Natl. Acad. Sci. USA*. **120**, e2300770120 (2023).